



# 1 Fracking bad language: Hydraulic fracturing and earthquake risks

*Jennifer J Roberts\*[1], Clare E. Bond[2], Zoe K. Shipton[1]*
1. Department of Civil and Environmental Engineering, James Weir Building, 75 Montrose St,
University of Strathclyde, Glasgow, G1 1XJ, Scotland, UK.
2. School of Geosciences, Department of Geology and Petroleum Geology, Meston Building, Aberdeen
University, Aberdeen, AB24 3UE, Scotland, UK
*corresponding author: jen.roberts@strath.ac.uk

## 8 Abstract

Hydraulic fracturing, fracking, is a well stimulation technique used to enhance permeability to aid
geological resource management, including the extraction of shale gas. The process of hydraulic fracturing
can induce seismicity and the risk of seismicity is a topic of widespread interest and is often reported to
be an issue of public concern regarding hydraulic fracturing. This is particularly the case in the UK, where
seismicity induced by hydraulic fracturing has halted shale gas operations, and triggered moratoria.
However, there seems a disconnect between the level of risk and concern around seismicity caused by
shale gas operations as perceived by publics and that reported by expert groups (from industry, policy,
and academia), which could manifest in the terminology used to describe the seismic events (tremors,
earthquakes, micro-earthquakes). In this paper, we examine the conclusions on induced seismicity and
hydraulic fracturing from expert-led public facing reports on shale gas published between 2012 and 2018
and the terminology used in these reports. We compare these to results from studies conducted in the
same time period that explore public views on hydraulic fracturing and seismicity. Further, we surveyed
participants at professional and public events on shale gas held throughout 2014 to elicit whether they
associate shale gas with earthquakes. We use the same question that was used in a series of surveys of
the UK publics in the period 2012 – 2016, but we asked our participants to provide the reasoning for the
answer they gave. By examining the rationale provided for their answers we find that an apparent
polarisation of views amongst experts is an artefact and in fact responses are confounded by ambiguity
of language around earthquake risk, magnitude, and scale. We find that different terms are used to
describe earthquakes, often in an attempt to express the magnitude, shaking, or risk presented by the
earthquake, but that these terms are poorly defined and ambiguous and do not translate into everyday
language usage. Such "fracking bad language" has led to challenges in the perception and communication
of risks around earthquakes and hydraulic fracturing, and leaves language susceptible to emotional
loading and misinterpretation. We call for multi-method approaches to understand perceived risks around
geoenergy resources, and suggest that adopting a shared language framework to describe earthquakes
would alleviate miscommunication and misperceptions. This work is relevant for a range of applications
where risks are challenging to conceptualise and poorly constrained; particularly those of public interest
where language inconsistency can exacerbate communication challenges and can have widespread
consequence.

## 37 1. Introduction

The pressing need for effective and acceptable decision-making on complex sociotechnical issues such as
climate change creates an unprecedented challenge. Effective dialogue between stakeholders, such as
scientists, policy makers and the publics, is crucial to tackle this challenge. However, communication can
be confounded by multiple issues, one of which is differences in understanding and use of language and
concepts.



The role of geoscience in modern sociotechnical issues faces particular challenges for several reasons.
First, geoscience underpins many issues of environmental and societal importance, such as resource
development (water, energy resources) and understanding and mitigation of climate change. These issues
are not only important for future generations, but associated activities (e.g. resource extraction,
development of low-carbon energy projects) have direct and indirect socio-economic and environmental
impacts at a range of scales (Leach, 1992; Vergara et al. 2013; Adgate et al., 2014, Stephenson et al., 2019).
Secondly, many geoscience concepts and technologies, as well as the geological resources that modern
lives depend on, are uncertain or unfamiliar to the wider public. This is complicated by the fact that the
Earth's subsurface is by nature both heterogenous and largely inaccessible. Amongst geoscientists such
uncertainty affects the confidence of predictions (Lark et al. 2014; Bond, 2015) and can lead to differing
interpretations (Bond et al. 2007; Alcalde et al., 2019) - even scientific dispute (compare interpretations
of the N. Sea Silver Pit Crater (Stewart and Allen, 2002; Stewart and Allen, 2004; Underhill, 2004). Thirdly,
the inaccessibility of and general unfamiliarity with the subsurface can make it challenging for lay publics
to conceptualise it (Gibson et al., 2016), and particularly to conceptualise geological processes or climate
and engineering risks (Taylor et al. 2014). Finally, geoscience terminology is often ambiguous,
incomprehensible for the public, or has multiple meanings. As an example, it is common to use ambiguous
phrases or descriptors such as 'deep' in the Earth, 'low levels' of contaminants, a 'large' fault, or 'geological
timescales'. Even the technical language used to describe geological observations can imply a specific
conceptual model or processes, or have slightly misleading meanings relating to the (since outdated)
origins of the word, and can lead to miscommunication amongst geoscience experts (Shipton et al., 2006;
Bond et al. 2007).
We posit that these socio-technical communication challenges may affect stakeholder's perception of the
efficacy of geological engineering approaches, and ultimately, their uptake. There are numerous
geoscience applications where apparent differences between expert and lay perspectives on technical
issues such as geological risk or environmental impact have affected development (Lowry, 2007; Vander
Becken et al., 2010; Scheider and Schneider, 2011; Graham et al., 2015; Marker, 2016). Hydraulic
fracturing (often referred to as 'fracking') for shale gas presents one such current and high-profile
example. Here, we explore the perception of and terminology around the perceived risks and
opportunities presented by hydraulic fracturing for shale gas in the UK context. In particular, we seek to
test whether technical expertise or familiarity with geoscience concepts influences the perceived risk of
induced seismicity (i.e. seismicity caused by human activity, in this case, by hydraulic fracturing or related
processes). This work is timely, as there is a clear need for further social scientific insights to inform risk
management and communication around geoenergy-induced seismicity (Trutnevyte & Ejderyan, 2018).
To frame our work, we first consider the importance of common or shared language as a communication
tool amongst stakeholders. We then provide a brief overview of shale gas exploration activities in the UK
and the associated socio-political ramifications before we review in detail the public and technical
discourse on induced seismicity. Doing so provides the context for the second part of our paper in which
we explore differences in the perceived risk of seismicity[1] and the language used to describe seismicity
associated with shale gas.
While this work is based on use of language in the communication and understanding of shale gas
extraction processes, the lessons are equally applicable to a range of geological engineering approaches

---

[1] We use the term seismicity in the body of the paper as a catchall term to describe a range of
phenomena that include: earthquakes, tremors, and so on. Secondly, although we focus on seismicity
in this paper, in doing so we do not construe any specific importance to this or other issues associated
with shale gas extraction. We merely use it as a pertinent example of the importance of language use
in scientific communication.





which may (be perceived to) present risk of induced seismicity (including hydropower dam construction,
nuclear waste disposal, carbon capture and storage, geothermal energy extraction, energy storage and so
forth), many of which are fundamental to delivering a sustainable future (Trutnevyte & Ejderyan, 2018;
Stephenson et al., 2019). Further, the learnings around language and communication, and inferences
about research methods are applicable to issues far beyond resource development.
*1.1 Language and communication in the geosciences*
Language is the method by which information, concepts and ideas are shared; it is a fundamental part of
being human (Heidegger, 1971). The audiences involved can range from the very small, i.e. between
individuals or small groups, to the very large, such as the global communication portal provided by the
world wide web. For researchers, communication approaches have changed with time. Oral forms of
scientific communication started with Victorian scientific debates (Yeo, 2003), evolved to become talks at
conferences and events, and is now broadening for example through live-streaming of events and using
channels such as online video content, and other modern creative approaches such as storytelling and
spoken word poetry (Dahlstrom, 2014; Brown, 2015). Written forms of scientific communication initially
took the form of books and monographs, but it is now primarily through peer-review publications (Banks,
2008, 2016), supplemented by affiliation-associated reports, policy briefs and blog articles.
There has been growing moves to increased public involvement in scientific issues - from funding
priorities, data collection, and policy decisions - particularly on topics with social and environmental
importance such as climate change, flooding, energy policy, genetically modified crops (e.g. Rowe et al.,
2005; Parkins and Mitchell, 2005; Horlick-Jones et al. 2007; Nisbet, 2009). This progression brings a new
communication challenge: for scientists, policy makers and the publics to be able to share information,
concepts and ideas, they must adopt a shared language or at least understand language translation. The
truth is that within languages, and here we are considering only the English language, there are sub-
sections that are only accessible to the learned few (i.e. those with technical expertise on the matter at
hand). Jargon is prevalent within science and underpins the explanation of many scientific concepts
between experts (Montgomery, 1989), but is in general incomprehensible to those outside the subject
area (Leggett and Finlay, 2001; Sharon and Baram-Tsabari, 2014). This creates an 'unequal communicative
relationship' whereby citizens struggle to comprehend the technical language and administrative goals
set by experts (Fischer, 2000, p. 18), particularly as many experts are ill-equipped to communicate with
members of the public (Simis et al., 2016).
This unequal relationship is likely enhanced in the geosciences where seemingly non-technical uncertain
or ambiguous terms are used routinely but are underpinned by some tacit understanding. Using depth
descriptors as an example, geoscientists may refer to 'shallow' earthquakes and a 'shallow' resistivity
survey, meaning a depth of several kilometres below surface for the former and meaning tens of metres
below the surface for the latter. But tacit understanding is not reliable; loose use of language, ambiguity
and poorly defined technical terms can lead to mis-understanding amongst experts (van Loon, 2000;
Doust, 2010).
It is well established that how individuals perceive new information is influenced by factors such as
expertise, context, prior knowledge, and the language used (McMahon et al., 2015; Venhuizen et al.,
2019). Consider the original work on framing by Tverskey and Kahneman (1981). In their example, when
disease treatment options were framed positively (lives saved) rather than negatively (lives lost) people
chose more risky treatment options. Similar work has found that how geoscience data and information is
framed affects decision making (Taylor et al., 1997; Barclay et al., 2011; Alcalde et al., 2017).
There has been a notable shift in the framing of positive and negative arguments around shale gas
extraction in the UK. Early arguments adopted local frames (i.e. concerns about local effects such as
induced seismicity, traffic, noise), and these arguments have tended to become replaced by global frames
(i.e. concerns about the climate change implications of developing onshore gas resources, Hilson, 2015,



or the role of natural gas in the energy transition, Partridge et al., 2017). Despite this, as we show in the
remainder of this section, issues around the risk of induced seismicity remain current, as exemplified by
the UK government decision on November 2$^{nd}$ 2019 to resuspend hydraulic fracturing for shale gas in
England and Wales.
In this work we are interested to explore whether and how different types of and levels of technical
expertise on hydraulic fracturing for shale gas development affects the perceived types of and levels of
associated risks, and the language used to describe these risks. As we have indicated, we focus specifically
on the risk of induced seismicity because this has been a common theme within the risk discourse on
hydraulic fracturing since shale gas development is an issue of public importance in the UK, and because
shale gas operations in the UK were repeatedly halted due to induced seismicity.
*1.2 Hydraulic fracturing, induced seismicity, and UK shale gas development*
In geological engineering contexts, hydraulic fracturing (often referred to as 'fracking') refers to the
process of fracturing rocks at depth by injecting pressurised fluids. The process locally increases the
permeability of the rock formation which is useful for a range of applications ranging from improving
water extraction, enhancing deep geothermal energy production, to enabling the recovery of 'tight gas',
where natural gas is trapped in rocks with a low permeability such as shales. Hydraulic fracturing occurs
in nature also, often where geological processes cause geofluids to become so overpressured that they
overcome the rock strength and cause the rock to fracture (e.g. Davies and Cartwright, 2007; Fall et al.,
149 2015).

As a rock fractures, seismic energy is released (e.g. Tang and Kaiser, 1998) as a seismic event, or seismicity.
For fracking, because the fracturing process is man-made, the seismicity is categorised as 'human-induced
seismicity' or, simply, 'induced seismicity'. Many other processes induce seismicity, from mining and
quarrying, to filling and dewatering reservoirs, to disposing of wastewaters by injecting them into deep
rock formations (Westaway & Younger et al., 2014; Pollyea et al., 2019). However not all seismic events
have any detectable effect in terms of being felt, or recorded.
There are a number of approaches to quantify, and so report on, the size of a seismic event. The moment
magnitude ($M_w$) relates to the seismic moment, which is the energy released by the slip surface. The local
magnitude ($M_L$) measures the ground displacement. The two scales $M_L$ and $M_W$ are fundamentally
different, and so the $M_w$ and $M_L$ of a seismic event can diverge, particularly for large (> $M_w$ 6.0) and small
(< $M_w$ 2.0) events (Clarke et al., 2019; Kendall et al., 2019). Seismologists prefer $M_w$ because it relates to
the properties of the fracture (the seismic moment) and because $M_L$ breaks down for smaller events
(below $M_L$ 2) (Kendall et al., 2019). However $M_L$ is easier to use for real-time reporting, and so is used to
report seismic events and to regulate induced seismicity (Butcher et al., 2017). A variety of terms are used
to describe any seismic event, including earthquakes, tremors, micro-earthquakes and so on.
Seismologists ascribe terminology based on the property of a seismic event, such as the frequency content
or the magnitude (for example, see Bonhoff et al., 2009), but there is no common classification
framework.
By definition, since any opening of a fracture will release energy and so induce seismicity, shale gas
extraction by hydraulic fracturing is not an exception. Put differently, any hydraulic fracturing will be
accompanied by release of seismic energy as the rock is fractured by the fluid pressure (Kendall et al,
2019). Induced seismicity should therefore be expected from hydraulic fracturing. But there are
uncertainties regarding the measurement, forecasting of and magnitude of these events (Kendall et al.,
2019). The nominal detection level for the UK network is $M_L$ = 2.0 (i.e. events above $M_L$ 2 might be felt at
the surface) (Kendall et al., 2019), whereas acoustic monitoring systems can record very small seismic
events down to magnitude $M_w$ -4 (e.g. in mines, see Kwiatek et al. 2011, Jalali et al., 2018). Whether or
not an event is felt at the surface depends on several factors, including the seismic moment, the
hypocentral depth and the attenuating properties and structure of the rocks through which the energy





travels, and other local conditions (Butcher et al., 2017; Kendall et al., 2019). Further, recorded $M_L$ is
dependent on the seismic detection network, including the array density and location distance between
source and detector (Butcher et al., 2017).
There are well documented incidences of felt seismicity associated with shale gas extraction from the US
(Warpinski et al. 2012), as well as from wastewater injection (Elsworth et al. 2015). However, the largest
recorded induced seismic events associated with shale gas extraction activities (not from associated waste
water disposal) have all occurred in the UK. In 2011, at the Preese Hall site in Lancashire (NW England)
induced a series of seismic events with maximum magnitude ($M_L$) 2.3 and 1.5 (Clarke et al., 2014),
suspending operations; and latterly in August 2019 a series of seismic events with maximum magnitude
2.9 $M_L$ was recorded at the Preston New Road site also in Lancashire.
These seismic events have led shale gas extraction to have the high public and political profile that it has
today, in the UK (Green et al. 2012; Selley, 2012; Clarke et al. 2014) and elsewhere, receiving widespread
media coverage and stimulating a wave of public protests against shale gas extraction (c.f Jaspal & Nerlich,
2014). The UK government introduced a moratorium on hydraulic fracturing activities 6 months following
the 2011 events. While the moratorium has remained in place in Scotland, it was lifted in England and
Wales in December 2012 with the introduction of new regulatory requirements intended to effectively
mitigate seismic risks (DECC, 2013a; DECC 2013b), known as the 'traffic light system' (Figure 1), based on
the local magnitude ($M_L$) of induced events. In November 2019 the moratorium was reapplied following
publication of the Oil and Gas Authority's report on the August 2019 earthquakes at Preston New Road
(BEIS, 2019a; OGA, 2019).
It is with this backdrop that we look to examine the available evidence of expert and non-expert
perspectives on the risks of seismicity associated with hydraulic fracturing, and the language and terms
adopted when describing these risks.

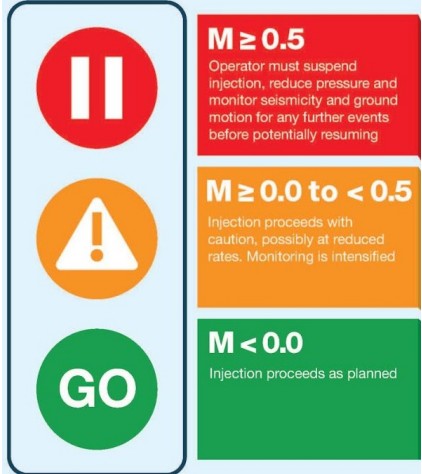


**Figure 1**: UK government 'traffic light system' for regulating induced seismicity from hydraulic fracturing activities
for shale gas extraction, figure from DECC (2013b), made by the Oil and Gas Authority. The traffic light system is
based on a risk mitigation technique originally developed for geothermal (Cremonese et al., 2015), and it requires
operators to monitor seismic activity in real time and if seismic events are detected, to proceed or stop depending
on the magnitude ($M_L$) of these events. Under this legislation, activities at Preston New Road were suspended several
times during hydraulic fracturing in December 2018 (OGA, 2019).





## 2. Induced seismicity and hydraulic fracturing: a review of perspectives and language used

In order to investigate expert and non-expert views, we must first define these terms. 'Expert' is a flexible term, but is usually applied to a person considered to be particularly knowledgeable or skilled in a certain field (Lightbody and Roberts, 2019). Here, we consider expertise to refer to in-depth knowledge about an aspect of the shale gas industry; be it technical (environmental regulation, geological, geological engineering, petroleum engineering, oil field services), or topical (energy policy and politics, energy or gas markets, regulation, environmental impact assessment, financing projects and investments). The wider publics or 'lay' audiences are not expected to have such expertise, they have lay-views, which we refer to here as non-expert. However, we recognise that publics can hold valuable experiential and contextual knowledge, rather than (but not excluding) technical or topical knowledge.

To examine expert and non-expert perspectives on induced seismicity we review publicly available resources (published before November 2019). For expert views, we look to reports from expert groups such as learned societies, expert panels and scientific enquiries. These materials draw on a range of evidence, including peer-reviewed publications in scientific journals, and are generally intended for a stakeholder audience, including the publics. We do not consider peer-reviewed publications in scientific journals, since relevant outcomes should be captured within the expert reports. For lay perspectives, we look to social science studies examining public opinions on hydraulic fracturing and shale gas development. While we are interested to examine non-expert views specifically about induced seismicity relating to hydraulic fracturing in the UK, the topic is usually referred to within resources on the broader context of shale gas exploration or development.

A summary of outcomes from expert-led publications are shown in Table 1A, and from studies of public perceptions around shale gas topics in Table 2.

### 2.1 Expert and lay perspectives on the risk of induced seismicity for hydraulic fracturing

Most expert reports conclude that the risks of induced seismicity from fracking in the UK are very low (Table 1). It is therefore fair to conclude that there is scientific consensus that the risks of induced seismicity are low, lower or no different to other human-induced seismicity, but also that these risks are sensitive to site and activity specific factors. To be clear, scientific consensus on induced seismicity does not reflect consensus on other aspects of shale gas debate, such as the business case for or environmental ethics of, fracking (Howell, 2018; Van de Graaf et al., 2018).

In contrast, studies of public perceptions (non-expert) around shale gas topics find a range of concerns around induced seismicity. These studies and their findings are summarised in Table 2. Table 2 also illustrates the similarities/differences in the phrases used in these studies, typically by researchers either in the design of the survey or process, or during analysis, to refer to induced seismicity. Some studies, and particularly online UK-wide surveys, report that publics view induced seismicity to be an important risk, and an indicator of public acceptability of shale gas (Andersson-Hudson et al., 2016; Howell, 2018). Other studies, particularly deliberative approaches, find that while risk of induced seismicity may be identified and raised as a concern, other perceived risks take higher priority (Whitmarsh et al., 2014; Thomas et al., 2017). To examine these insights in more detail, we first summarise insights from cross-public (i.e. nationwide) surveys before we look to more qualitative approaches. In each case, mindful that public views may have evolved in response to shale gas activities in the UK, the studies are presented chronologically in the order in which they were conducted (not the order in which they were published). Where possible we also report results of the terminology adopted in each study (e.g. earthquake vs seismicity vs tremor). As before, we are interested in the perspectives of induced seismicity, and not the public opinion around fracking more generally.

A number of surveys have been undertaken to assess UK-wide public attitudes towards shale gas and related topics. The most comprehensive of these in terms of a longitudinal dataset is the YouGov survey

organised by University of Nottingham. The survey was administered 12 times in the period March 2012
- October 2016 (Andersson-Hudson et al., 2016; O'Hara et al., 2016). Following a knowledge question
which filtered out participants that don't know what hydraulic fracturing or shale gas is, respondents were
then asked questions about multiple aspects of shale gas development. One question asked whether they
do or do not associate earthquakes with shale gas, with the option to answer 'don't know'. In the period
2012-2014, there is a steady decline in the number of participants who associate shale gas extraction with
earthquakes (Figure 2), which remains around ~50%, and a corresponding increase in those that do not.
The surveys also gathered participants' levels of support for shale gas, and repeatedly find that individuals
who do not associate shale gas with earthquakes are more likely to support shale gas extraction, and vice
versa.

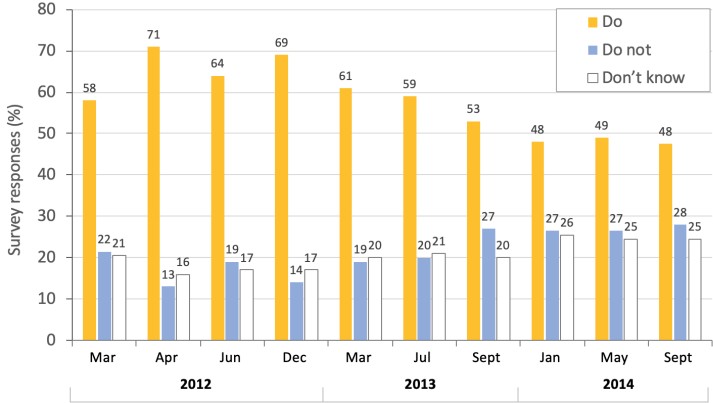

"Do you associate earthquakes with shale gas?"


**Figure 2**: Responses to the University of Nottingham surveys administered via YouGov to assess public perspectives
on shale gas development for the ten surveys between 2012-14 (c.f. O'Hara et al., 2016). The number of participants
that associate shale gas with earthquakes decreases, while the number of participants that do not associate, or don't
know, increases.

Other UK-wide surveys that have regularly canvassed public attitudes towards shale gas include the
Energy and Climate Change Public Attitudes Tracker. The survey is conducted quarterly by the Department
of Energy and Climate Change (now the Department of Business, Energy and Industrial Strategy; BEIS) to
capture changing public attitudes towards energy and climate change issues. Questions about shale gas
have been included in the survey since June 2012, and since 2015, reasons for support, opposition, or no
view have been enquired about (Howell, 2018). One of the frequent reasons for opposition to shale gas
that is consistent across the BEIS surveys includes concern around the risk of earthquakes (Bradshaw &
Waite, 2017). Opinium Research led two online surveys to explore public attitudes to fracking in 2014 and
2015 (reported in Howell, 2018). The survey did not ask participants about perceived risks. However,
questions from the Opinium Research were adapted for a different online omnibus survey fielded by
YouGov, also 2015 (Howell, 2018). Howell (2018) found the majority (43.2%) of respondents who
answered a knowledge question about shale gas correctly agreed that "fracking could cause earthquakes
and tremors", whereas 18.8% disagreed (the remainder answered 'don't know'). However, the level of
positive response for earthquakes and tremors ranked towards the lowest of the range of negative
environmental and social risks (including damage to the local environment, water contamination, negative
affect on climate change, and health risks). A one-off online survey in 2014 (Whitmarsh et al., 2015) finds
that 40.4% of participants agreed that they are "concerned about the risks of earthquakes from shale gas



fracking", with half that (20.8%) reporting that they disagreed, and the remainder undecided, but public
were less concerned about earthquakes than they were about water contamination.
The most recently published survey, UK National Survey of Public Attitudes Towards Shale Gas, conducted
in April 2019, is the first to seek to understand what the public knows or thinks about specific regulations
for shale gas, including the 'traffic light system' for monitoring and regulating induced seismicity (Evensen
et al., 2019). The majority of participants feel that the traffic light guidance is not stringent enough, and
would oppose any changes to raise the threshold to 1.5 $M_L$, suggesting that concerns around induced
seismicity from hydraulic fracturing remain (Evensen et al., 2019).
Overall, these surveys indicate that induced seismicity is an important issue for publics. However, many
of these surveys asked closed questions, and so to some degree the topics of concern are pre-identified
during the survey design, and are shaped by the phrasing question. For example, the Whitmarsh et al.
(2015) survey asked questions in the style "I am concerned about [environmental risk]"; other questions
in the same survey were focused on risks around energy security or energy prices, and did not use the
words 'concern' or 'risk', both of which have negative associations. Similarly, Howell (2018) found the
question, "fracking could cause earthquakes and tremors", is interpreted to be a negative statement
about fracking, rather than a factual statement. The results of these closed surveys should therefore be
interpreted and compared with some caution.
Indeed, while qualitative analysis of data presented in the public inquiry on planning permission for shale
gas development in Lancashire (held in 2016) found that "*seismic activity was raised regularly in the public
sessions. Several of those who spoke had first-hand experience of seismic activity having felt the tremors
from Cuadrilla's hydraulic fracturing at Preese Hall in 2011*" (Bradshaw & Waite, 2017), findings from other
qualitative research suggest that such concerns be relatively low importance compared to other perceived
risks. For example, Craig et al. (2019) studied public views towards fracking and how these changed with
distance from a region of County Fermanagh with potential shale gas resources and a granted petroleum
exploration license. Survey results, which were gathered in 2014, indicated that risk of 'increased
seismicity' ranked eighth amongst the ten common risks considered to be a concern by survey
respondents. All of the identified risks increased with proximity to the licensing area, including the
perceived risk of increased seismicity due to hydraulic fracturing.
Williams et al. (2017) reports on deliberative focus group discussions on shale gas development. The
groups were held in north England in 2013, and the results suggest that explicit concern about induced
seismicity was not expressed, although some groups did express 'worst case scenario' thinking around a
number of potential risk and impact pathways (Williams et al., 2017). Similarly, a series of 1-day
deliberations in the UK and the US held in 2014 found that participants did not express particular concern
about induced seismicity (Thomas et al., 2017). In deliberative interviews held in Wales in 2013/14 the
risk of earthquakes or tremors was ranked 13[th] out of 19 pre-identified risks (Whitmarsh et al., 2014). In
2016 a citizens' Jury (a format for public deliberation) was held in Preston, Lancashire (NW England)
approximately 10 miles from the Preese Hall shale gas development. Transcriptions from the proceedings
show that while participants raise questions around earthquake risks from shale gas extraction (and
geological $CO_2$ storage), concerns about induced seismicity were not a dominant issue (Bryant, 2016).
Our review indicates that the reported level of public concern about induced seismicity suggested by the
results from UK-wide surveys may be a product of the survey structure, including the phrasing of, or the
type of questions that are asked and also a product of the analysis and reporting of survey results.
Deliberative and dialogic approaches find that concerns around the risk of induced seismicity are not as
significant as the surveys suggest; while concerns around induced seismicity are raised, it is not a primary
or dominant concern within the context of other perceived risks. That said, Thomas et al. (2017) report
that deliberative groups in the UK and the US felt that if shale development were to cause earthquakes,
however small, development should not be pursued. Similarly, Williams et al. (2017) reports how one
deliberative group reflected that public tolerances to industrial activities (which induce seismicity) may





have changed such that activities that were acceptable in the past are no longer acceptable to the public.
Finally, early results from a recent investigation into public attitudes to the UK governments traffic light
system to regulate induced seismicity suggest that the public support stringent monitoring of induced
seismicity. These insights imply that the perceived risk presented by seismicity from hydraulic fracturing
activities may not be acceptable to the publics, despite the level of concern being relatively low compared
to other perceived risks. This evidence for a continued perceived risk has been born-out by the
reintroduction of a moratorium on fracking in the UK in November 2019.
*2.2 Language used by expert and lay audiences on the risk of induced seismicity*
Experts use a range of terms to describe induced seismicity (Table 1). The seismic events themselves might
be referred to as (*micro-seismic events*, *seismicity*, and *earthquakes*. A distinction is made between
*natural* and *induced* earthquakes, and the events that may occur from hydraulic fracturing or other man-
made activities are described as being *induced* by or *triggered* by these activities (where induced can mean
solely due to fracking, and triggered can mean that the occurrence was accelerated by fracking, but might
have occurred naturally). Qualifiers such as *minor*, *low-magnitude*, *small* are frequently used to indicate
the size or magnitude of seismicity associated with fracking. Finally, while the consequences of seismicity
are sometimes referred to in terms of *vibrations* or *tremors*, more often there is a distinction between *felt*
and *not felt* events.
In some cases, the language around seismicity in policy reports is misleading, inconsistent and confusing.
For example, in a DECC (2013) report, the new regulatory requirements are first described as designed
"to ensure that seismic risks are effectively mitigated" (p6). However, the regulations are later described
as designed "to prevent any more earthquakes being triggered by fracking" (p19), despite induced seismic
events of magnitude ($M_L$) < 0.5 being permissible ("green light") according to those regulations. On the
next page (p20) an additional qualifier is added which gets around this contradiction: the regulations are
"designed to prevent any more *perceptible* earthquakes being triggered by fracturing". A recent OGA
report (which summarised a series of studies commissioned by the OGA to understand and learn from the
induced seismicity observed at Preston New Road developments in 2018) concluded that rules based on
current understanding of induced seismicity cannot be "reliably applied to eliminate or mitigate induced
seismicity" (OGA, 2019). The authors do not define what is meant by induced seismicity.
In comparison, the terminology to describe induced seismicity reported in public perception studies is
much less varied (Table 2). In many cases, the phrases are selected by the researchers, either when
designing the survey question or when reporting on the research outcomes. For example, four of the five
closed question surveys with about induced seismicity refer to risk of '*earthquakes*'. Results from the only
survey to add a size-qualifier, asking about '*earthquakes or tremors*' (Howell, 2018), are very similar to the
results of surveys which simply asked about '*earthquakes*'. In contrast, of the phrasing chosen by
researchers (to report on results from the remaining two surveys which asked open-ended question about
perceived risks and shale gas, or to report on the results from deliberative approaches), only one study
refers to '*earthquakes*' (Thomas et al., 2017). Researchers prefer to use terms such as '*(increased) seismic
activity*', '*seismicity*', or '*minor earthquakes*'. The phrases that publics themselves may adopt are not
reported in these studies, except for in the report on the citizens' jury on fracking where, in their
questions, participants wanted to get to grips with whether the 2011 Preese Hall seismic events had been
"*real/genuine*" earthquakes (caused by hydraulic fracturing) or "*natural tremor*" (i.e. background
seismicity) (Bryant et al., 2016, pp 14).
*2.3 Knowledge, language and perceived risks of induced seismicity*
As Jaspal and Nerlich (2014) reflect, terms such as 'earthquakes' evoke imagery of destruction and
disaster, whereas phrases like 'seismic activity' or 'tremors' are less threatening. The distinction in
language used in the survey questions and the language used to summarise qualitative discussions on the



perceived risks might be telling about the level of risk perceived by the publics, in line with results from deliberative research approaches.

Further, since hydraulic fracturing, by definition, will induce (albeit small) seismic events, it could be argued that assertions such as "shale gas development is associated with earthquakes" are factual, and do not indicate the level of perceived risk. Indeed, results from the Howell (2018) survey show that respondents who correctly answer a knowledge question about shale gas are more likely to agree with the statement "fracking could cause earthquakes and tremors" (43.2%) than to answer don't know (38.0%) or to disagree (18.8%). While Howell (2018) report no significant difference in the overall level of support for fracking in the UK between those who evidenced knowledge and those who did not, this contrasts with results from the University of Nottingham surveys (Andersson-Hudson et al., 2016) who find no association between knowledge and support for shale gas. Further, results from these surveys repeatedly suggest that whether or not respondents associate shale gas with earthquakes correlates to their support for shale gas development (Andersson-Hudson et al., 2016; 2018).

In summary, through our review and analysis of previous surveys, reports and papers, we have revealed uncertainties in the perceived risk of seismicity induced by hydraulic fracturing for shale gas. There is broad scientific consensus amongst experts that induced seismicity may be associated with hydraulic fracturing, the likelihood of felt seismicity is dependent on context specific technical factors, but the risk presented by such seismicity is low. In contrast, evidence on the perceived risk of induced seismicity amongst lay publics is mixed. UK-wide surveys indicate that publics associate fracking with earthquakes, and often this correlates with, or is inferred to indicate, opposition to fracking. However, deliberative approaches present a more nuanced perspective, whereby any level of induced seismicity may be deemed to be unacceptable, even though perceived risks associated with shale gas development are often considered to be more concerning than induced seismicity. Further, the language used by experts to refer to induced seismicity is much more varied than the language used to report on public views on the matter.

In the next section, we explore whether or not knowledge levels affect whether seismicity is associated with shale gas, and how the language used in the questions asked affects the answer provided.




| Year | Report (*purpose*) | Conclusion on (risk of) induced seismicity | Terminology used to describe seismicity |
|---|---|---|---|
| 2012 | **Mair et al. (2012)** Royal Society and Royal Academy of Engineering (2012) 'Shale gas extraction in the UK: a review of hydraulic fracturing' *Report commissioned by UK Government Chief Scientific Adviser.* | "Seismic events induced by hydraulic fracturing … do not produce ground shaking that will damage buildings. The number of people who feel small seismic events is dependent on the background noise." (pp 16) "Magnitude 3 ML may be a realistic upper limit for seismicity induced by hydraulic fracturing (Green et al. 2012)" (pp 41). The report recommends a traffic light system to be put in place (transferred learning from geothermal energy developments) | Varied terminology, including: *induced seismicity, seismic event, vibrations, felt/not felt, magnitude* and *intensity.* |
| | **AEA (2012)** AEA Report for European Commission DG Environment 'Identification of Potential Risks for the Environment and Human Health arising from Hydrocarbons Operations involving Hydraulic Fracturing in Europe' *Report commissioned by the European Commission DG Environment to inform policy.* | The risk of significant induced seismic activity was considered to be low; the frequency of significant seismic events is judged to be "rare" and the potential significance of this impact is "slight" (pp 60) | Tend only to refer to *very small magnitude, seismic activity, earth tremors.* |
| | **Green, C. A., et al. (2012)** Preese Hall shale gas fracturing review and recommendations for induced seismic mitigation. *Report commissioned by DECC to examine the possible causes of seismicity at Preese Hall in April/May 2011.* | The report concludes that the observed seismicity in April and May 2011 was induced by the hydraulic fracture treatments at Preese Hall. The authors also conclude that, providing that proposed best practice operational guidelines are implemented and followed, the risk of induced seismicity should not prevent further hydraulic fracture operations in this area. | The authors primarily refer to *earthquakes* or *seismic events*, and sometimes refer to "*small*" events/earthquakes. |
| | **Kavalov & Pelletier (2012)** European Commission Joint Research Centre (2012) 'Shale Gas for Europe - Main Environmental and Social Considerations' *Undertaken by the European Commission's in-house science service to provide evidence-based scientific support to the European policy-making process.* | "Drilling and hydraulic fracturing activities may lead to low-magnitude earthquakes" (pp 26). The authors make no conclusions on risk, but recommend that "the severity and probability of this hazard should be carefully assessed on site by site basis". | Refer only to *low-magnitude earthquakes* |
| 2013 | **DECC (2013c)** DECC Report 'About shale gas and hydraulic fracturing' *Government response to common questions raised in the UK-wide consultation on shale gas and fracking.* | Regulations are designed to "ensure that seismic risks are effectively mitigated". | A mix of terms are used, including *seismicity, events, activity, tremors.* The most frequent term is *earthquake*, in some cases with qualifiers such as *perceptible, large, small, very small.* |



| | | | |
|---|---|---|---|
| | **National Research Council (2013)** US National Research Council 'Induced Seismicity Potential in Energy Technologies' | "The process of hydraulic fracturing a well as presently implemented for shale gas recovery does not pose a high risk for inducing felt seismic events" (pp 18). | Only refer to *earthquakes* and *seismicity* |
| | **Cook et al. (2013)** Australian Council of Learned Academies (ACOLA) Unconventional Gas Production: A study of shale gas in Australia *Report the Prime Minister's Science, Engineering and Innovation Council* | Induced seismicity from hydraulic fracturing itself does not pose a high safety risk (pp 137). Risks can be managed by adopting a range of mitigation steps. | *Earthquakes* or *seismicity* are used most often, but with qualifiers such as *minor, low magnitude, felt*. |
| **2014** | **European Commission (2014)** European Commission Recommendation on minimum principles for the exploration and production of hydrocarbons using high-volume hydraulic fracturing *EU Regulation/legislation* | The recommendations refer only to risk assessment protocols for induced seismicity, not the risk of earthquakes per se. | Refers only to *seismicity* |
| | **Scottish Government (2014)** Expert Scientific Panel on Unconventional Oil and Gas Development *Report from an expert panel set up by Scottish Government* | "seismic effects are expected to be small in magnitude" (pp 39); "very low likelihood of felt seismicity" from fracking (pp 48) | A number phrases are used. S*eismicity* is often pre- by *micro-, trigger/induce,* or *felt*. Also refer to *tremors, (natural) earthquake*. |
| **2015** | **TFSG (2015)** Task Force on Shale Gas 'Assessing the Impact of Shale Gas on the Local Environment and Health' *Second report by the industry-funded expert panel Task Force on Shale Gas.* | "Shale gas operations have the potential to cause tremors albeit not at a level higher than …other comparable industries in the UK, nor at a frequency or magnitude significantly higher than natural UK earthquakes" (pp 9). | Refer mostly to *earthquakes* and *tremors* (and to a lesser extent, '*events*'), but often prefacing these terms with words such as *small, tiny, minor, micro*. |
| | **Cremonese et al. (2015)** Institute for Advanced Sustainability Studies (IASS) Potsdam Policy Brief Shale Gas and Fracking in Europe *Policy brief to inform European Policy* | Site-specific stress investigation will reduce risk of seismicity (pp 3). | Refer to *small* induced *seismic events* |
| **2016** | **Baptie et al. (2016)** Unconventional Oil and Gas Development: Understanding and Monitoring Induced Seismic Activity. *Report commissioned by Scottish Government* | Hydraulic fracturing to recover hydrocarbons is generally accompanied by earthquakes with magnitudes of less than 2 ML that are too small to be felt. (pp 2). | Only refer to *earthquakes* and *seismicity or seismic activity*, but often specify that these events are induced. Sometimes refer to *felt*. |
| **2018** | **Scottish Government (2018)** Report for Scottish Government's SEA on unconventional gas *Report commissioned by Scottish Government* | The risk of fracking-induced felt seismicity causing damage to properties or people at the surface is considered to be very low. Risk table reports that felt seismic activity would have minor negative or negligible effect on activities. | Range of terms including felt *seismicity, earthquakes, trigger* |
| | **Delebarre et al. (2018)** House of Lords Briefing paper CBP 6073 'Shale gas and fracking' | No position indicated - but quote several expert reports which state that the risk of induced seismicity can be managed. | *Seismicity* is used most frequently. *Earthquakes* and *events* also commonly used. *Tremor* and *trigger* used infrequently. |



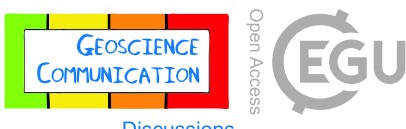

| | | | |
|---|---|---|---|
| | *Briefing paper to inform House of Lords debate.* | | |
| **2019** | **BEIS (2019b)** Guidance on fracking: developing shale gas in the UK (updated 12 March 2019) *UK Govt Department for Business, Energy, and Industrial Strategy* | "The UK's strong regulatory regime is ensuring hydraulic fracturing only happens in a safe and environmentally responsible way." "Measures are in place to mitigate seismic activity." | *Seismicity* or *seismic activity* are most often referred to. Do not refer to *earthquakes*. |
| | **OGA (2019)** Oil and Gas Authority 'Interim report of the scientific analysis of data gathered from Cuadrilla's operations at Preston New Road' *Summary outcomes from four reports commissioned by OGA in response to induced seismicity at Preston New Road.* | It is currently not possible to "reliably eliminate or mitigate induced seismicity" (pp 13). | *Seismicity* is most often used, with some reference to *events* and *activity*. |

**Table 1**: A compilation of publicly available expert reports on hydraulic fracturing for shale gas which address
induced seismicity, the key conclusion regarding risks of induced seismicity and the phrasing used in the reports to
refer to seismicity. While we primarily examine policy-facing reports from the UK, we include examples from EU
policy, Australia and the US.

| | **Source** | **Year data collected (method)** | **Findings on public perception of induced seismicity** | **Phrases adopted (by who)** |
|---|---|---|---|---|
| **Surveys** | Andersson-Hudson et al. (2016) | 2014 (University of Nottingham YouGov survey; sample size: 3,822) | Whether or not *earthquakes* are associated with hydraulic fracturing is an indicator of support for shale gas | Earthquake (researchers designing the survey question) |
| | Craig et al. (2019) | 2014 (face-to-face surveys in four locations; total sample size: 120) | Risk of *increased seismicity* was ranked 8 out of 10 identified risks associated with fracking | Increased seismic activity (researchers phrasing in their analysis) |
| | Evensen (2017) | 2014 (University of Nottingham YouGov survey; sample size: 3,823 + US survey, sample size: 1,625) | UK public associated *earthquakes* with shale gas more than US publics | Earthquake (researchers designing the survey question) |
| | Whitmarsh et al. (2015) | 2014 (local/regional online survey; sample size: 1,457) | When asked if they were concerned about the risks of *earthquakes* from shale gas fracking, 40.4% agreed and 20.8% disagreed | Earthquakes (researcher's phrasing in their survey design) |
| | Howell (2018) | 2015 (YouGov online omnibus survey; sample size: 1,745) | Fracking could cause *earthquakes and tremors* (43.2% agree, 18.8% disagree) | Earthquakes or tremors (researchers designing the survey question) |



|  |  |  |  |  |
|---|---|---|---|---|
|  | Andersson-Hudson et al. (2019) | 2016 (University of Nottingham YouGov survey; sample size: 4,992) | Whether or not *earthquakes* are associated with hydraulic fracturing is an indicator of support for shale gas, particularly for more knowledgeable participants | Earthquake (researchers designing the survey question) |
|  | McNally et al. (2018) | 2017 (face-to-face surveys in one location; sample size: 200) | *Seismicity* was raised as a primary concern when the phrase "fracking" was used in the survey question | Seismicity (researcher's phrasing in their analysis) |
|  | Evensen et al. (2019) | 2019 (YouGov online survey; sample size: 2,777) | Some level of concern around the risks of *seismic activity* is implicit in the public attitudes towards the traffic light system (which is perceived not to be stringent enough) |  |
| **Deliberative approaches** | Whitmarsh et al. (2014) | 2013-2014 (deliberative interviews, sample size: 30; local/regional survey, sample size: 1,457) | Deliberative interviews: minor earthquakes were not considered to be a principle risk associated with hydraulic fracturing; they ranked 13th out of a pre-defined list of 19 risks). Surveys: 40.4% agreed, 20.8% disagreed that they were "concerned about the risks of earthquakes from shale gas fracking". | Minor earthquakes or earthquakes (researcher's phrasing in their survey design) |
|  | Williams et al. (2017) | 2013 (six deliberative focus groups; total sample size: 48) | Explicit concern about induced seismicity wasn't expressed | Seismicity (researcher's phrasing in their analysis) |
|  | Thomas et al. (2017) | 2014 (Series of four 1-day deliberative workshops, two in UK, two in the US; total sample size: 55) | Some concerns were raised regarding earthquake risk, but these weren't particularly important in the context of the deliberations. However, all four groups felt that if shale development were to cause earthquakes, however small, shale gas should not be pursued at all. | Earthquakes (researcher's phrasing in their analysis) |
|  | Bradshaw & Waite (2017) | 2016 (qualitative analysis of a public enquiry into shale gas in Lancashire, UK; sample size: N/A) | Concerns about seismic activity were voiced by publics during the inquiry proceedings. | Seismic activity (researchers' phrasing in the paper) |
|  | Bryant (2016) | 2016 (citizens jury in Lancashire; sample size: 15) | Questions about seismic activity were asked, but concerns about induced seismicity wasn't explicitly mentioned in the deliberation outcomes. | "real" or "genuine" earthquake, "natural tremor", as referred to by participants. |




**Table 2**: A compilation of published studies which report on public perceptions of induced seismicity in the UK. These
are divided into surveys (many of them UK-wide) and more qualitative approaches such as focus groups, and each
group is ordered chronologically in terms of when the data were gathered (not in terms of when the papers were
published). We identified whether the phrasing used (to describe seismic events) was dictated by the language of
the survey questions, or the researcher undertaking the analyses, or the participants themselves.



**3. A survey to examine the rationale and language use behind perspectives on induced seismicity and hydraulic fracturing**

**3.1 Methodology**

*3.1.1 Data collection*

We recruited 387 participants from a series of geoscience events on shale gas that were held in 2014, including conferences and public talks (see Table 3). We invited attendees to voluntarily complete and return the surveys, which were anonymous. Our sample includes 204 participants from shale gas specific conferences, 85 participants from geoscience conferences (that were not shale gas specific), and 98 participants from science outreach events[2] on shale gas. Since a number of individuals attended several of the conferences and events we requested that people only complete the survey once.

| Acronym | Event name (location; date) | Description | N (surveys) |
|---|---|---|---|
| **Shale gas specific events** | | | |
| ESGOS | European Shale Gas and Oil Summit (London; 09/2014) | An industry led conference on shale gas | 40 |
| UGA | Unconventional Gas (Aberdeen; 03/2014) | An industry led conference on shale gas | 28 |
| SGUK | Shale Gas UK (London; 03/2014) | An industry led conference on shale gas | 98 |
| **Geoscience events** | | | |
| TSG | Tectonic Studies Group Annual Conference (Cardiff; 01/2014) | The annual conference of the Geological Society of London specialist group covers a range of topics relevant to tectonic studies. The event included a technical session on hydraulic fracturing and induced seismicity, followed by an open discussion. | 57 |
| CCG | Communicating Contested Geoscience (London; 06/2014) | A Geological Society of London conference about issues facing controversial geoscience topics, including shale gas. | 66 |
| **Public events** | | | |
| TFA | TechFest (Aberdeen; 09/2014) | Talk and discussion at a local science festival | 30 |
| CSA | Café Science (Aberdeen; 02/2014) | Talk and discussion at a Café Science, a popular science communication series that occur across the UK. | 59 |
| CHL | Coffee House Lectures (Glasgow; 11/2014) | Talk and discussion at a local research communication series | 9 |

**Table 3**: The events where attendees were invited to anonymously complete surveys. Public events were generally small local events.

*3.1.2 Survey design*

We adapted a subset of questions from the University of Nottingham surveys (O'Hara et al. 2014; Andersson-Hudson et al., 2016). The questions we used were intended to gather information on the perceived risks of and level of support for shale gas development, and asked for closed answers to a series

---

[2] These events lasted between 1-2 hrs and consisted of an interactive talk (by one or more of the authors of this paper) followed by a discussion session. All three talks were part of small local events held in Scotland.





of statements about shale gas. Crucially, in our modified survey, participants were asked to provide
reasoning for the answers they gave.
Conference participants were asked to report which sector they worked in, and all participants were asked
to report their sources of information about or experience of shale gas (as a proxy for their maximum
knowledge-level on the topic).
Full survey data (raw and analysed) are available at <insert DOI when generated>.

*3.1.3 Data Analysis*

In this work, we consider only the responses to the closed question "*please state whether you do or do
not associate earthquakes with shale gas*" (from which respondent could select either 'do', 'do not', or
'don't know') and the open question seeking the reasoning behind the selected answer to the closed
question. In total 385 participants completed the closed question (99% of survey respondents), and 292
participants provided informative responses to the open question (67.5% of survey respondents).
Closed answers were coded numerically. Open answers were categorised through thematic coding to
enable analysis. The codes for thematic analysis were derived iteratively as follows: Firstly, the three
authors of this paper worked separately on open coding (i.e. inducing themes from the qualitative answers
to all questions). We then had a series of collective workshops (between the three of us) to share
identified codes, determine similarities or differences in our codes, and then discuss and refine the
identified themes until they were reconciled and consolidated, and both the themes and their definition
or scope agreed. The authors then worked separately again to apply the codes across all qualitative
answers (in several cases a single answer was double or treble coded). The lead author then co-ordinated
the codes, seeking consensus in the few cases of disagreement between the applied codes.
Thematic analysis of all qualitative data (reasoning provided for the selected answer to the closed survey
question about earthquakes) derived a total of 26 themes, of which 15 apply to answers about induced
seismicity. These are shown in Table 4. Qualitative answers were coded as null if the content was
irrelevant, i.e. did not explain the rationale for the answer provided (the most common example being a
knowledge statement about the topic, for example, "I've analysed this issue", "I work on this topic") or
the meaning of the response was ambiguous and couldn't be deciphered. Overall 80% of all respondents
provided qualitative responses that were thematically coded.
We examine how these themes vary with job sector and knowledge level. Employment sector responses
were grouped into academia, industry, civil service, and other. Most of the 289 conference participants
who completed the survey were from industry (52%) and academia (30%), with only 12% from the civil
service (3% did not answer this question). Information sources on the topic of shale gas were grouped
into no prior information, information from media reports, expert reports, and academic research (95%
of survey respondents answered this question). We consider individuals whose knowledge sources
include reports and academic papers to be highly informed (i.e. experts). The majority (81%) of the
conference attendees were in this knowledge category, with 40% obtaining information from academic
papers and 41% from reports. In contrast most (60%) public talk attendees sourced information about
shale gas from media.
The public cohort were not intended to represent the perspectives of the general public. The surveys were
completed at the end of a public talk and discussion on the topic of shale gas, in which induced seismicity
was raised, and so these publics are both interested and informed, and therefore cannot be a proxy for
UK-wide attitudes and responses. Instead, the public cohort allow us to examine answers for those who
obtain the majority of prior information, if any, through media sources (most conference attendees do
not fit this category). Public respondents were not asked about employment sector.
We compare results from our survey with those from the 12 University of Nottingham YouGov surveys
(O'Hara et al., 2016). While the Nottingham YouGov surveys document a broad decline in the number of
respondents that associate shale gas with earthquakes (see Figure 2), the results for the three surveys
undertaken in 2014, the period in which we undertook our surveys, do not show any decline. We use
average values from 2014 surveys (48% do, 27% do not, and 25% don't know) to represent UK-wide views,



against which we compare our results. For simplicity, we refer to these as the '*UoN 2014*' surveys and
results.

| Code | Description: The reason provided indicates that…. | Dir. |
|---|---|---|
| **Evidence** | There is evidence that shale gas extraction [causes/induces/is associated with] earthquakes.<br>*Includes references to events in the USA. References to UK events are coded as below.* | ↑ |
| **Blackpool** | Any reference to the seismic sequences at Preese Hall in 2011 as evidence of risk of earthquakes.<br>*Includes references to Lancashire, Blackpool, Cuadrilla or more broadly to UK events.* | ↑ |
| **Inconclusive** | There is currently not enough evidence to (conclusively) say whether or not shale gas extraction [causes/ induces/is associated with] earthquakes.<br>*Includes reference to a need for further research/data (to understand the positive and negative impacts, to improve technology and so on)* | ↔ |
| **No evidence** | Shale gas extraction is not associated with [do not cause or induce / is associated with] earthquakes. | ↓ |
| **Knowledge** | Respondent doesn't feel that they know enough about shale gas extraction to say. Or they are on the fence. | ↔ |
| **Media** | Reference to the media coverage of shale gas extraction.<br>Phrases include: *press, news, high profile, reporting, public concern, miscommunication, scaremongering, hype, anti-fracking activist, anti- lobby.* | ↑ |
| **Fracturing rock** | Shale gas extraction requires the reservoir rock to be hydraulically fractured. This process will release seismic energy.<br>Phrases include: *inherent/obvious, fracturing rock, high-pressure fluids, stress change, trigger.* | ↑ |
| **Waste-water** | Shale gas extraction may not induce earthquakes, but the geological disposal of waste-water (associated with fracking) does.<br>Phrases include: *waste water, waste disposal/injection, USA events.* | ↑ |
| **Reactivation** | There is a risk that shale gas extraction may cause earthquakes because the process may reactivate existing fractures and faults which could cause seismicity | ↑ |
| **Magnitude** | The magnitude of any seismic events related to fracking will be very small.<br>Phrases include: *micro (seismic/earthquake), tremor, low intensity/energy, tiny, cannot feel them, insignificant, low consequence/impact* | ↓ |
| **Low risk** | The risk that shale gas extraction [causes/induces/is linked with] earthquakes is very low.<br>Phrases include: *is possible, rare, unlikely, low risk, minor, little impact, not a significant risk.* | ↓ |
| **Definition** | Comments or questions how earthquake is defined. | ↔ |
| **Regulation** | The risk that shale gas extraction activities may cause earthquakes can be managed by appropriate regulation and monitoring. Includes reference to regulation, appropriate regulation, enforcing regulation, best practice.<br>Phrases include: *monitoring, controllable, manageable* | ↓ |



| Normal | Any seismic activity that may be induced by shale gas extraction is no different to everyday/background/other activities or industries. i.e. not unique to fracking. | ↓ |
|---|---|---|
| Site | Any risk posed by shale gas extraction is location or place specific. Phrases include: *determined by the geology of the region, the depth of the resource, the population etc.* | ↔ |

**Table 4:** Codes identified for thematic analysis of participant responses to an open question asking them to provide
reasoning for the answer they gave to the closed question. The codes are often directional, i.e. they are used to
reason why earthquakes may be associated with shale gas (positive ↑), why earthquakes may not be associated
with shale gas (negative ↓). If the code is not directional (or it is bi-directional) it is considered to be neutral (↔).





**3.2 Survey Results and Analysis**

*3.2.1 Closed question responses*

In total 55% of survey respondents who answered the closed question ("*do you associate shale gas with earthquakes"*) '*do*' associate shale gas with earthquakes, 37% '*do not*' and 7% '*don't know*' (Figure 3A). Compared to public attitude surveys asking the same question throughout 2014, our survey finds more respondents '*do*' (+7%) '*do not*' (+10%) and far fewer '*don't know*' (-18%). Overall our respondents are much more decided than the general public (see Figure 2, O'Hara et al., 2016). Of our cohort, we find more participants from professional fora such as conferences and events (which are about, or have sessions about, shale gas) '*do*' associate shale gas with earthquakes (58%) than participants attending public talks (48%) (Figure 3B).

We observe no obvious trend between the closed answer responses and participant knowledge levels (expertise), but we do observe differences (Figure 3C). When grouped into experts and non-expert groups (those who source information from research and reports, and those who had no prior information or obtained information from the media, respectively), 56% of experts (n. 276) associate shale gas with earthquakes and 39% do not. These proportions are very similar to non-experts (n. 109) where 53% do and 33% do not, and are in fact very similar to the views of UK-wide publics in 2013, see Figure 2. However, grouping in this way masks a difference in responses between those who obtain information from research articles and those who use reports. For the latter, shale gas is predominantly associated with earthquakes, (64% do; 31% do not) whereas for the former, there is a fairly even split (49% do; 47% do not) (Figure 3C). These experts are not undecided, their views are polarised.

The only group that predominantly do not associate shale gas with earthquakes are those with no prior knowledge of shale gas, although this sample is very small (n. 16). Our results present a more nuanced view than the results of Andersson-Hudson et al. (2016) which find that those with more knowledge about shale gas are more likely not to associate shale gas with earthquakes.

It would be fair to presume that most academics would source their information from research papers, and so it is interesting that the results for job sector present a different perspective (Figure 3D). Two response profiles emerge from job sector results: academics and civil service workers (where 65% (academics) 68% (civil service) associate earthquakes with shale gas; 28% (academics) 21% (civil service) do not), and industry, who present an even mix of views (51% do; 46% do not), similar to those that obtain information from research articles.

*3.2.2 Open question responses*

Thematic analysis of open responses (which provided reasoning for participants' closed answer to the question '*do you associate shale gas with earthquakes'*) identify 15 codes, which are shown in Table 5 (the thematic codes definitions are listed in Table 4). Often multiple codes apply to a given answer, and so in total, there are 443 codes for the 292 qualifying responses. Codes are ranked for frequency in Table 5. The six most frequently used codes are identified over 30 times in participant responses, and these themes are examined in more detail in Table 6.

Themes relating to *magnitude* were most often raised in participant responses, and accounted for over a quarter of the total number of codes applied across all open responses (Table 5), inclusive of knowledge level or job sector (Table 6) and 40% of the open responses. The code is equally prevalent across reasoning to support '*do*' and '*do not*' responses, but less frequent for *'don't know'* answers (where unsurprisingly *inconclusive* and *knowledge* themes become important even though the sample is very small).

The *magnitude* theme illuminates uncertainty in what is understood to be an earthquake, and raises questions around terminology. This is best illustrated using example answers from this theme, shown in Table 7. Thus, the same reasoning is being provided to support different closed answers. Other common codes include *low risk* and *media*. The *low risk* theme provides similar reasoning to *magnitude* but refers to risk rather than scale of the event (Table 7), and the reasoning is provided to all perspectives ('*do*', '*do not*', '*don't know*'). In contrast, *media* is used mostly to describe reasons for answering '*do*', alongside





reference to the Blackpool (Preese Hall) seismic events, and the rationale that *fracturing rock* inevitably
releases seismic energy and so fracking and earthquakes are associated by definition. Where the *media*
theme is used for '*do not*' responses, often the respondent is expressing judgement about the accuracy
or veracity of media claims.

**Figure 3 (A)** Comparing the results of our surveys with UK-wide results from 2014 (UoN 2014; O'Hara 2015), we
find that while results for 'do' associate shale gas with earthquakes (orange) for both surveys are similar our survey
results have more 'do not' (blue) and much fewer 'don't know' answers (grey).
**(B):** Participants from professional fora (conferences and events, pale green) associate earthquakes with shale gas
more than participants from public talks on shale gas (green). Results are compared to UK-wide results from 2014
(UoN 2014; O'Hara 2015) (dark green).
**(C):** To gauge knowledge levels of our survey participants, we asked respondents to select where they source their
information from about shale gas, with 'research papers' indicating the greatest knowledge and 'no previous
information' indicating the least prior knowledge. There is no overall trend to the results, suggesting that answers
are not simply determined by knowledge level. In fact, those who obtain information from research present an
~equally polarised response, which is different to information from reports and the media where the dominant
answer is that earthquakes are associated with shale gas. The only group to report that shale gas is not associated
with earthquakes is the small sample of respondents that obtained no information about shale gas prior to
attending the event where they completed the survey.
**(D):** The majority (83%) of participants recruited at conferences and events (n. 272) source from industry and
academia (public participants were not asked their job sector). We observe some differences in closed question
responses between the different sectors; while the majority of participants from academia, the civil service and
other sectors predominantly report that earthquakes are associated with shale gas, industry participants are



almost 50:50 do and do not associate shale gas with earthquakes. Very few of those from industry and academia
(~5%) answer don't know.
Two additional themes are identified in the rationale for '*do not*' responses. First, the argument that any
earthquakes associated with shale gas extraction will be no more significant than other everyday
background seismicity or industry processes, and so is considered to be *normal*. This code is unique that
it is used mostly to support *do not* responses. Further, in their reasoning for '*do not*' responses, a number
of participants raise questions about how the term earthquake is *defined*. Themes around earthquake
*definition* also arise within rationale for '*don't know*' responses (Table 7), with the same questions being
raised regardless of the answer: '*what is the difference between microseismic event and an earthquake?*'.
Some respondents confidently assert that microseismic events or tremors are not earthquakes, others
indicate that earthquakes refer to 'natural' seismic events (similar to comments made by Citizens Jury
participants, as reported in Bryant, 2016).
Results presented in Table 6 indicate that neither knowledge level or job sector have any significant
influence on the themes raised in open responses. We observe only two small trends; participants from
industry tend to appeal to *media* themes more than other sectors, and academics are more likely to refer
to *Blackpool* events (i.e. the Preese Hall events) as an indicator that earthquakes are associated with shale
gas development.






| | Evidence | Blackpool | Inconclusive | No evidence | Knowledge | Media | Fracturing rock | Waste-water | Reactivation | Magnitude | Low risk | Definition | Regulation | Normal | Site |
|---|---|---|---|---|---|---|---|---|---|---|---|---|---|---|---|
| Do | 7 (3%) | 30 (11%) | 1 (0%) | 1 (0%) | 1 (0%) | 32 (12%) | 29 (11%) | 15 (6%) | 9 (3%) | 76 (28%) | 34 (13%) | 7 (3%) | 10 (4%) | 11 (4%) | 7 (3%) |
| Do Not | 2 (1%) | 3 (2%) | 2 (1%) | 5 (4%) | 0 (0%) | 9 (6%) | 6 (4%) | 8 (6%) | 2 (1%) | 38 (27%) | 18 (13%) | 16 (11%) | 6 (4%) | 21 (15%) | 5 (4%) |
| Don't Know | 0 (0%) | 1 (4%) | 5 (20%) | 0 (0%) | 5 (20%) | 3 (12%) | 0 (0%) | 0 (0%) | 0 (0%) | 3 (12%) | 4 (16%) | 3 (12%) | 1 (4%) | 0 (0%) | 0 (0%) |
| **Total** | 9 (2%) | 34 (8%) | 8 (2%) | 6 (1%) | 6 (1%) | 44 (10%) | 35 (8%) | 23 (5%) | 11 (3%) | 117 (27%) | 56 (13%) | 26 (6%) | 17 (4%) | 32 (7%) | 12 (3%) |
| Rank | 12 | 5 | 13 | 15 | 15 | 3 | 4 | 8 | 11 | 1 | 2 | 7 | 9 | 6 | 10 |


**Table 5**: The frequency of use of different thematic codes in the reasoning provided for participants' answers,
showing total number of times the code was applied and, in brackets, the percentage relative to the number of
responses in that category (do, do not, don't know). High frequency codes are coloured pale yellow (≥10%) and
yellow (≥20%). One answer (reasoning) could have more than one code. At the bottom of the table codes are
ranked for frequency, and the eight codes that occur over 20 times are coloured in blue. These themes are
examined in detail in Table 6.

| | | Magnitude ↓ | | | | Low risk ↓ | | | | Media ↑ | | | | Frac rock ↑ | | | | Blackpool ↑ | | | | Normal ↓ | | | |
|---|---|---|---|---|---|---|---|---|---|---|---|---|---|---|---|---|---|---|---|---|---|---|---|---|---|
| | | - | M | R | A | - | M | R | A | - | M | R | A | - | M | R | A | - | M | R | A | - | M | R | A |
| Do | n | 0 | 17 | 32 | 27 | 0 | 6 | 14 | 15 | 3 | 17 | 8 | 5 | 0 | 5 | 15 | 9 | 0 | 5 | 12 | 13 | 0 | 2 | 2 | 7 |
| | % | 0% | 15% | 27% | 23% | 0% | 10% | 24% | 26% | 7% | 37% | 17% | 11% | 0% | 14% | 41% | 24% | 0% | 15% | 35% | 38% | 0% | 6% | 6% | 22% |
| Do Not | n | 2 | 5 | 16 | 15 | 3 | 0 | 4 | 11 | 0 | 2 | 5 | 3 | 0 | 0 | 0 | 7 | 0 | 1 | 0 | 2 | 0 | 8 | 4 | 9 |
| | % | 2% | 4% | 14% | 13% | 5% | 0% | 7% | 19% | 0% | 4% | 11% | 7% | 0% | 0% | 0% | 19% | 0% | 3% | 0% | 6% | 0% | 25% | 13% | 28% |
| Don't Know | n | 0 | 1 | 1 | 1 | 0 | 2 | 1 | 2 | 1 | 0 | 1 | 1 | 0 | 0 | 0 | 1 | 0 | 1 | 0 | 0 | 0 | 0 | 0 | 0 |
| | % | 0% | 1% | 1% | 1% | 0% | 3% | 2% | 3% | 2% | 0% | 2% | 2% | 0% | 0% | 0% | 3% | 0% | 3% | 0% | 0% | 0% | 0% | 0% | 0% |
| Sum | n | 2 | 23 | 49 | 43 | 3 | 8 | 19 | 28 | 4 | 19 | 14 | 9 | 0 | 5 | 15 | 17 | 0 | 7 | 12 | 15 | 0 | 10 | 6 | 16 |
| | % | 2% | 20% | 42% | 37% | 5% | 14% | 33% | 48% | 9% | 41% | 30% | 20% | 0% | 14% | 41% | 46% | 0% | 21% | 35% | 44% | 0% | 31% | 19% | 50% |


| | | Magnitude ↓ | | | | Low risk ↓ | | | | Media ↑ | | | | Frac rock ↑ | | | | Blackpool ↑ | | | | Normal ↓ | | | |
|---|---|---|---|---|---|---|---|---|---|---|---|---|---|---|---|---|---|---|---|---|---|---|---|---|---|
| | | A | I | CS | O | A | I | CS | O | A | I | CS | O | A | I | CS | O | A | I | CS | O | A | I | CS | O |
| Do | n | 25 | 29 | 10 | 2 | 7 | 12 | 6 | 2 | 4 | 13 | 0 | 0 | 10 | 13 | 1 | 2 | 11 | 8 | 2 | 2 | 3 | 2 | 4 | 1 |
| | % | 26% | 30% | 10% | 2% | 16% | 28% | 14% | 5% | 15% | 50% | 0% | 0% | 29% | 38% | 3% | 6% | 44% | 32% | 8% | 8% | 12% | 8% | 16% | 4% |
| Do Not | n | 7 | 17 | 2 | 1 | 1 | 11 | 1 | 0 | 1 | 5 | 1 | 0 | 2 | 5 | 0 | 0 | 0 | 2 | 0 | 0 | 4 | 10 | 0 | 1 |
| | % | 7% | 18% | 2% | 1% | 2% | 26% | 2% | 0% | 4% | 19% | 4% | 0% | 6% | 15% | 0% | 0% | 0% | 8% | 0% | 0% | 16% | 40% | 0% | 4% |
| Don't Know | n | 1 | 0 | 1 | 1 | 1 | 0 | 1 | 1 | 0 | 1 | 1 | 0 | 0 | 0 | 1 | 0 | 0 | 0 | 0 | 0 | 0 | 0 | 0 | 0 |
| | % | 1% | 0% | 1% | 1% | 2% | 0% | 2% | 2% | 0% | 4% | 4% | 0% | 0% | 0% | 3% | 0% | 0% | 0% | 0% | 0% | 0% | 0% | 0% | 0% |
| Sum | n | 33 | 46 | 13 | 4 | 9 | 23 | 8 | 3 | 5 | 19 | 2 | 0 | 12 | 18 | 2 | 2 | 11 | 10 | 2 | 2 | 7 | 12 | 4 | 2 |
| | % | 34% | 48% | 14% | 4% | 21% | 53% | 19% | 7% | 19% | 73% | 8% | 0% | 35% | 53% | 6% | 6% | 44% | 40% | 8% | 8% | 28% | 48% | 16% | 8% |


| <10% | 10 - 25% | 25 - 40% | >40% |
|---|---|---|---|

**Table 6:** Code frequency and (**A**) different information sources (for all participants) and (**B**) employment sector (for
conference attendees) for the six most frequent codes (organised from left to right in order of code frequency).
Information sources range from no information source (-); media (M); reports (R); (A) research (academic) papers,
and where employment sector for conference participants: Academia (A); Industry (I); Civil Service (CS), and other
(O). The count for each code is normalised to the total count for that code. These values are then colour coded as
shown in the key to indicate where codes are used by particular knowledge or employment groups, or to support
particular answers.






| | Closed response | Example open response (quotes) |
|---|---|---|
| **Magnitude** | *Do* | "the earthquakes associated with shale gas are very small", will be "microseismic earthquakes that won't be felt", "small magnitude events" or "minor tremors". |
| | *Don't know* | "major earthquakes probably unlikely", fracking may cause "seismic activity, but not quakes". |
| | *Do not* | "there may be possible tremors - not earthquakes", "events will be "mostly unfelt, very small events", or that there a "very few cases [with] little intensity". |
| **Low risk** | *Do* | Shale gas "can trigger earthquakes but very rarely", "has the potential to induce seismic activity, but the risk is not a significant" and "any induced seismicity [has] small consequences". |
| | *Don't know* | "It is probably unlikely that fracking triggers major earthquakes", there is "probably an association but the risk is relatively trivial" and earthquakes might be associated "with a tiny minority of shale [operations, they are] not an intrinsic by product". |
| | *Do not* | "Seismicity risks are minimal and manageable" "insignificant", "very low", "unimportant", and so "don't consider it [to be] a significant hazard". |
| **Media** | *Do* | Earthquakes are associated with shale gas due to "publicity", "media reports" "media portrayal and local campaign group resources". Responses also include judgement statements such as "thanks to the media I associate fracking with [earthquakes], but I don't agree". |
| | *Don't know* | "media and other bias form of reporting on shale gas give this impression however I don't know of any evidence of the link". |
| | *Do not* | "'*Earthquakes*' are associated publicly with shale gas thanks to inaccurate media reporting", "while I don't [associate shale gas with earthquakes], from media alone I would do". |
| **Normal** | *Do* | "We have a lot of evidence of earth tremors associated [with shale gas], but these are…comparable to historic mining activity in the UK" |
| | *Do not* | "Earthquakes can be induced from many different types of industrial processes", "numerous unfelt earthquakes occur daily, and [there are] only a select few examples of fracking caused felt earthquakes", "any earthquakes from shale gas will be negligible versus natural seismicity". |
| **Definition** | *Do* | "Fracking causes microseismicity, in rare occasions they cause earthquakes. Where is the transition between microseismic [events] and earthquakes?" Fracking does "create microseismicity… not on the scale you would call an earthquake". "Earth tremors or seismic events is more appropriate than earthquake". |
| | *Don't know* | Fracking might cause "tremors but not specifically earthquakes". "I think of earthquakes' as being of natural origin" |
| | *Do not* | "I don't think the minor, largely insensible tremors associated with shale gas merit the term 'earthquake'." "Seismicity" "tremors" "microseismicity" "is not an earthquake". |

**Table 7**: Example open response to illustrate how the most common codes are used to defend the range of
participant responses to whether or not they associate shale gas with earthquakes. *Magnitude* is generally used to
defend do and do not answers, *risks* is used for all responses, whereas *media* most often applies to 'do' answers.
*Normal* and *definition* codes tend to be applied to *do not* answers.
*3.2.3 Language and terminology*
A theme that is applied in particular to the rationale for '*do not*' answers refers to the definitions of
earthquakes, indicating that different phrases are more appropriate depending on the scale, size or
magnitude of the seismic event. We examine the language used within participants' open responses to
examine whether there are any language preferences amongst different answers or different survey
groups.



Participants used a range of terms to describe or refer to earthquakes. Similar words are used to describe
earthquakes in responses for both '*do'* and '*do not'* closed answers, though there is some indication that
words like *seismic* and *tremor* are used more for '*do not'* responses. The only distinction in terminology is
that more knowledgeable participants (experts - those that obtain information from reports and peer-
review publications) are four times more likely to use phrases such as '*seismicity'* and '*minor'* than less
knowledgeable respondents (non-experts). Academics use the phrase earthquake far more than those
employed in other sectors, and civil service employees prefer '*tremor'* rather than '*micro'* or '*induced'*
seismicity, and more often refer to '*energy'* of the event.
Finally, an undercurrent theme to the open responses was to critique the question that they were asked,
which was about perceived association between shale gas and earthquakes. As noted in the previous
section, many participants raised questions about the phrase '*earthquake'*, claiming it was a '*too strong'*,
and that any seismicity that might arise from shale gas development would not be '*earthquakes'* but
'*tremors'* or '*micro-earthquakes'*. Others preferred to mention earthquake consequences in terms of felt
or not-felt, or damage-inducing or not. Several participants critique the use of the phrase 'shale gas',
mentioning that they did not associate *shale gas* with seismicity, but they do associate *the hydraulic*
*fracturing technique* (by which shale gas is extracted) with seismicity. Others note that the question is
leading. Finally, most of the respondents that raised themes relating to the code *low risk* were essentially
communicating that whether they '*do'* or '*do not'* associate shale gas and earthquakes, it does not concern
or worry them (see Table 7). These statements suggest that the assumption that associating shale gas
with earthquakes is to express concern about the risk of earthquakes is erroneous.



### 4. Discussion

The results from our survey reflect a snapshot of participant views from 2014 about induced seismicity and hydraulic fracturing. The results were not intended to inform whether or not earthquakes are associated with shale gas, but, rather, to explore the underlying rationale for the apparent differences in perspectives on the topic, particularly between experts and non-experts. It is important to acknowledge that perspectives may now differ even more, particularly given the repeated suspension of hydraulic fracturing activities in Lancashire due to induced seismicity. Preston New Road is the only shale gas hydraulic fracturing activity in Europe that has been undertaken since our surveys were conducted in 2014; many countries including Scotland had moratoria in place during this period, and, once the moratorium in England was lifted in 2012, it took several years to obtain planning permissions to enable activities to commence at the Preston New Road site. We cannot postulate whether the rationale for the answers provided by participants might have changed in light of these developments. Further, our results show perspectives from the UK only, a country with low background seismic activity; and for English language use. Nonetheless, our results do shed light on the ambiguity in the language around induced seismicity and the confusion that this can cause, the differences between publics and expert views on the matter (and difficulties in assessing expertise), and the limitations of using close surveys to elicit views on risk.

Expertise is an ambiguous quality with multiple dimensions that can be difficult to assess (c.f. Lightbody and Roberts, 2019). Many of our survey respondents were attending professional fora about shale gas, and therefore might be considered to have expertise on the topic. Those who attended public lectures on hydraulic fracturing could be said to be informed (and engaged) publics. Accordingly, we find that our survey participants are, on the whole, much more decided on the topic than the UK general public (based on the University of Nottingham surveys as reported in O'Hara et al., 2016). Of the relatively few participants in our survey who answered '*don't know*', their response did not necessarily reflect lack of knowledge; several explained that the evidence was inconclusive or questioned the definition of earthquake. Survey respondents who attended public events and who answered '*don't know*' were more like to express that they lack knowledge on the topic, and so we could conjecture that this is the likely rationale when UK publics' answer '*don't know*'. A fourth closed answer category '*undecided*' would capture these differences.

While fewer '*don't know*' responses might be expected of those working in shale gas topics or attending public lectures on shale gas, it is interesting that there remains *no consensus* amongst our survey respondents about whether or not earthquakes are associated with shale gas. While we find that the proportions of those who '*do*' associate earthquakes with shale gas vary according to different factors including the fora being attended (professional or public), the sources of information used to obtain information about shale gas (beyond the event they were attending, expert reports vs academic papers vs media) and job sector (academic, industry, civil service); in every case the results are bimodal. While this might be interpreted to show polarisation of views both amongst experts and publics, by examining the underlying rationale for the answers provided by our participants, we find this not to be the case. Participant answers are muddied by ambiguity of language which leads to differences in understanding of what defines or constitutes an earthquake, and what is meant by 'associating' earthquakes with shale gas. As a result, many participants attempt to communicate risk within their responses, too. Alongside the ambiguous definition of the term earthquake (particularly regarding the size of an event), the term 'associate' was felt by many respondents to be too lose. Some argued that it is possible to associate an event with a cause in media reporting of an event without any there being a scientific explanation for a causal process. As a result, many participants attempt to communicate their understanding of the causal process and hence the risk that an activity will result in an event within their responses, too.

Regardless of whether our respondents '*do*' or '*do not*' associate earthquakes with shale gas, qualitative answers most commonly express uncertainty around what magnitude of seismic event is understood to be an earthquake. In particular, those who '*do not*' associate earthquakes and shale gas question the definition of an earthquakes. The term *earthquake* (the phrase used in the survey question) is clearly felt to be ambiguous by our survey respondents. This aligns to similar language expressed by experts



interviewed by Lampkin (2018), in which one expert expressed "*I would call them tremors not*
*earthquakes, they are very very small*" and another asserts that "*people who talk of earthquakes are sort*
*of over-egging* [over doing] *it a bit*" (Lampkin, 2018).
So, what constitutes an earthquake? Is it wrong or, indeed '*over-egging it*' to describe a $M_L$ < 2 event as
an earthquake? Technically, not (Kendall et al., 2019). In which case, how should earthquakes be
described? There are multiple scales with which to describe the size or properties of earthquakes,
including different scales of magnitude and energy release. However, there is no common descriptive
scale to define whether an event is a tremor, a micro-earthquake, small or large, or felt. Tremor has
traditionally referred to low-frequency earthquake signals (Shelly et al., 2007), and terms such as micro-
or nano- seismicity often refer to the frequencies of the seismic energy. The degree to which an
earthquake is felt is captured by the European Macroseismic Scale, which includes classifications such as
not *felt*, *scarcely felt*, *weak*, *largely observed*, and Bohnhoff (2009) summarise terminology based on
magnitude, including *micro*, *small*, *moderate*, *large*. The UK Government's traffic light system infographic
(Figure 1, made by the Oil and Gas Authority) describes seismicity as *not felt*, *usually not felt*, *minor*, *light*,
*moderate, strong, major, great*. In our study, we have not encountered any consistence use of such
language when describing and reporting hydraulic fracturing seismicity in public or expert fora.
Our findings show that there is no common descriptive scale for earthquakes, and certainly none that
translate into common language and understanding, even among experts. We find that while expert
reports commonly refer to '*earthquakes*', '*seismicity*' and '*events*', many use additional qualifiers to
communicate the scale of the event by using terms such as '*small*' or '*tiny*', distinguishing between '*felt*'
or '*perceived*' events, or by referring to the consequences of the seismicity using terms such '*tremors*' or
'*vibrations*' (Table 7). Importantly, none of the reports that we reviewed lay out what is meant by these
different phrases, though some stipulate that felt seismicity is generally considered to be above $M_L$ 2.
Similarly, our survey respondents include indicators of size, risk, and impacts in their qualitative answers.
They might select that they '*do*' associate shale gas with earthquakes, but explain that '*any induced
seismicity would be small or rare*', or they may select that they '*do not*' associate shale gas with
earthquakes, because '*any induced seismicity would be small or rare*' (see Table 7). Thus whether or not
a respondent associates shale gas with earthquakes does not reflect the perceived risk of seismicity. We
posit that in the survey preamble, had we presented a definition of what was meant by the term
earthquake (e.g. the release of seismic energy, or seismic events with magnitude greater than 2 $M_L$) the
answers to the closed question would have been in much greater agreement.
These findings raise crucial questions around what constitutes an earthquake and to whom; and how
language is used to describe and communicate geological phenomena. A second important aspect that
our work highlights is the need to apply caution when using results or conclusions from surveys and
reports that do not define ambiguous terminology .
Previous studies have inferred that associating shale gas with earthquakes reflects the perceived risk of
seismicity. However, by examining the reasoning provided by participants to explain their responses, we
find that in reality this is much more nuanced amongst experts, and thus public concern about risks of
induced seismicity may not be as high as the results of previous surveys have been used to imply. Indeed,
our review finds that other studies of public attitudes indicate that while there is evidence of public
concern around induced seismicity, it may be that closed surveys with few questions or options conflate
the level of concerns. Indeed, large proportions of all survey participants are undecided, and (qualitative
approaches in particular find that) often other potential negative impacts associated with hydraulic
fracturing are considered to be more important than seismicity (see Table 2). It is important to note
however that that low levels of concern do not mean that the risk of induced seismicity is acceptable for
publics, as indicated by findings from public deliberations (Williams et al., 2017), and implied by results
from surveys which consistently finds that respondents who associate fracking with seismicity are less
likely to support shale gas (Andersson-Hudson et al., 2016, see Table 2).
These outcomes simply highlight the limitations of closed questions in surveys. Such questions are by their
nature constrained in scope, and so findings from closed questions are susceptible to bias and
simplification. Further, it is well documented that the framing of questions can affect the result; indeed



Howell (2018) proposes that differences in results between her work on shale gas perceptions and the
work of Andersson-Hudson et al. (2016) might be because the survey design was significantly different.
But altogether this raises important questions around the methods used to capture and communicate
stakeholder perspectives. Only when a multi or mixed method approach in which qualitative and
quantitative methods are combined can a comprehensive understanding of complex topics be obtained.
Unlike the UK regulations, public risk tolerances of induced seismicity will not simply relate to event
magnitude; there are other complicating factors at play (Trutnevyte & Ejderyan, 2018; Szolucha, 2019).
Our findings about language, ambiguity, and potential to conflate perceived risks have application across
a range of different geological and energy engineering technologies, many of which play a critical role in
delivering a sustainable future (Stephenson et al., 2019). We propose that a shared language to describe
earthquakes should be developed and adopted to enhance communication around induced seismicity
amongst all stakeholders. Such approach is common in risk communication and management practice
(Fischhoff, 2013), and has recently been called for by a community of UK shale gas researchers and
practitioners (Brown et al., 2020). It supports communication, and, as put by Trutnevyte & Ejderyan
(2018), without such framework experts must develop their communication approaches based on
intuition and learning by doing [author note: these experiences are often described by practitioners as
being 'at the coal face' or 'on the front line', indicating the challenging pressured environment for
learning]. As noted previously, frameworks exist (such as the European Macroseismic Scale; Johnston,
1990; Bohnhoff, 2009, and so on) but have not been adopted and translated into common language use.
While a common language framework would facilitate risk communication, it would not resolve
communication and risk tolerance challenges around induced seismicity. Any risk communication strategy
must be individual to project, place and context, as well as sensitive to issues of environmental and social
equity and justice and heritage in which geoenergy is embroiled (Trutnevyte & Ejderyan, 2018), and the
risk presented by some technologies may be more acceptable than others (Knoblauch et al., 2018).
However the framework would establish a common understanding through language, which is critical for
dialogue on topics of public and political interest, and could have mitigated the miscommunication,
misperception, and misinterpretation documented in this work.

**5. Conclusions**

This work has explored expert and non-expert perspectives around the risk of induced seismicity from
shale gas exploration in the UK. We find that range of terminologies have been inconsistently used to
describe seismic events to communicate risk of induced seismicity, and we highlight how language
ambiguity and question framing has muddled understanding of the perceived risk of induced seismicity
and hydraulic fracturing. Our insights present important implications for research, communication, and
decision-making on any uncertain, complex or sensitive topic. The immediate and long-lasting
repercussions of using "fracking bad language" is likely amplified by the political and environmental
sensitivities around the shale gas sector, as well as lack of familiarity of seismicity (natural and induced)
to UK stakeholders. We suggest that a shared language to describe earthquakes should be developed and
adopted to facilitate risk communication within and between expert and non-expert stakeholders. This
framework will be relevant for numerous geoscience applications, where many subsurface technologies
deemed critical to a low carbon future are unfamiliar to the publics and present risk of induced seismicity.
Finally, our work illustrates the value of examining social scientific issues through a multi method lens to
inform risk management and communication.

**6. Data Availability**

Survey data are available at <insert DOI when generated>.



**7. Funding statement**

We thank ClimateXChange and the University of Strathclyde who funded Roberts' position while this research was undertaken.

**8. Ethics statement**

This research complied with the Ethics Policy and Procedure of the University of Strathclyde. Ethics approval was granted for the survey research.

**9. Competing interests**

We declare no competing interests.

**10. Author contributions**

JR lead the research design, data collection, analysis, and writing of this research, with CB in particular and ZS contributing to all aspects.

**11. Acknowledgements**

We thank all conference and event organisers for supporting our work, as well as survey participants. We also thank Dr Stella Pytharouli, Dr James Verdon, and Dr Stephen Hicks for their insights into earthquake magnitudes and seismological terminology, and Dr Juan Alcalde for comments about language nuance and translation. We would also like to thank Prof Brigitte Nerlich for early discussion about the relevance of this work.

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

A Literature Review European Commission Joint Research Centre Institute for Environment and
Sustainability.



Kendall, J. M., A. Butcher, A. Stork, J. Verdon, R. Luckett and B. J Baptie (2019). "How big is a small
earthquake? Challenges in determining microseismic magnitudes." First Break 37: 51-56.
Knoblauch, T. A. K., M. Stauffacher and E. Trutnevyte (2018). "Communicating Low-Probability High-
Consequence Risk, Uncertainty and Expert Confidence: Induced Seismicity of Deep Geothermal Energy
and Shale Gas." Risk Analysis 38(4): 694-709.
Kwiatek,G. et al. (2011) Source Parameters of Picoseismicity Recorded at Mponeng Deep Gold Mine,
South Africa: Implications for Scaling Relations Bulletin of the Seismological Society of
America(2011),101(6):2592 http://dx.doi.org/10.1785/0120110094
Lampkin, J. A. (2018). Will Unconventional, Horizontal, Hydraulic Fracturing for Shale Gas Production
Purposes Create Environmental Harm in the United Kingdom? Doctor of Philosophy, University of Lincoln.
Lark, R.M., Thorpe, S., Kessler, H. and Mathers, S.J., 2014. Interpretative modelling of a geological cross
section from boreholes: sources of uncertainty and their quantification. Solid Earth, 5(2), pp.1189-1203.
Leach, G., 1992. The energy transition. Energy policy, 20(2), pp.116-123.
Lowry, D., 2007. Nuclear waste: The protracted debate in the UK. In Nuclear or Not? (pp. 115-131).
Palgrave Macmillan, London.
Leggett, M. and Finlay, M., 2001. Science, story, and image: a new approach to crossing the
communication barrier posed by scientific jargon. Public understanding of science, 10(2), pp.157-171.
Lightbody, R. and J. J. Roberts (2019). Experts: The Politics of Evidence and Expertise in Democratic
Innovation. The Handbook of Democratic Innovation and Governance. S. Elstub and O. Escobar, Edward
Elgar Publishing.
Mair, R., M. Bickle, D. Goodman, B. Koppelman, J. Roberts, R. Selley, Z. Shipton, H. Thomas, A. Walker and
E. Woods (2012). Shale gas extraction in the UK: a review of hydraulic fracturing, Royal Society and Royal
Academy of Engineering.
Marker, B. R. (2016). "Urban planning: the geoscience input." Geological Society, London, Engineering
Geology Special Publications 27(1): 35.
McNally, H., P. Howley and M. Cotton (2018). "Public perceptions of shale gas in the UK: framing effects
and decision heuristics." Energy, Ecology and Environment 3(6): 305-316.
McMahon, R., Stauffacher, M. and Knutti, R., 2015. The unseen uncertainties in climate change: reviewing
comprehension of an IPCC scenario graph. Climatic change, 133(2), pp.141-154.
Montgomery, S.L., 1989. The cult of jargon: Reflections on language in science. Science as Culture, 1(6),
pp.42-77.
National Research Council. 2013. Induced Seismicity Potential in Energy Technologies. Washington, DC:
The National Academies Press. https://doi.org/10.17226/13355.
Nisbet, M.C., 2009. Framing science: A new paradigm in public engagement. In Communicating
science (pp. 54-81). Routledge.
O'Hara, S., M. Humphrey, J. Andersson-Hudson and W. Knight (2016). Public Perception of Shale Gas
Extraction in the UK: From Positive to Negative.
OGA (2019) Interim report of the scientific analysis of data gathered from Cuadrilla's operations at Preston
New Road. Available at: https://www.ogauthority.co.uk/media/6149/summary-of-pnr1z-interim-
reports.pdf
Parkins, J.R. and Mitchell, R.E., 2005. Public participation as public debate: a deliberative turn in natural
resource management. Society and natural resources, 18(6), pp.529-540.
Partridge, T., M. Thomas, B. H. Harthorn, N. Pidgeon, A. Hasell, L. Stevenson and C. Enders (2017). "Seeing
futures now: Emergent US and UK views on shale development, climate change and energy systems."
Global Environmental Change 42: 1-12.
Pollyea, R. M., M. C. Chapman, R. S. Jayne and H. Wu (2019). "High density oilfield wastewater disposal
causes deeper, stronger, and more persistent earthquakes." Nature Communications 10(1): 3077.





Rowe, G., Horlick-Jones, T., Walls, J. and Pidgeon, N., 2005. Difficulties in evaluating public engagement
initiatives: reflections on an evaluation of the UK GM Nation? public debate about transgenic crops. Public
Understanding of Science, 14(4), pp.331-352.
Schneider and Schneider (2011)Sharon, A.J. and Baram-Tsabari, A., 2014. Measuring mumbo jumbo: A
preliminary quantification of the use of jargon in science communication. Public Understanding of
Science, 23(5), pp.528-546.
Scottish Government (2014) Expert Scientific Panel on Unconventional Oil and Gas report ISBN:
974  9781784126834

Scottish Government (2018) Unconventional oil and gas policy: SEA ISBN: 9781787813014
Selley, R. C. (2012). "UK shale gas: The story so far." Marine and Petroleum Geology 31(1): 100-109.
Shelly, D., Beroza, G. & Ide, S. Non-volcanic tremor and low-frequency earthquake
swarms. Nature 446, 305–307 (2007). https://doi.org/10.1038/nature05666
Shipton, Z.K., Evans, J.P., Abercrombie, R.E. and Brodsky, E.E. (2013). The Missing Sinks: Slip Localization
in Faults, Damage Zones, and the Seismic Energy Budget. In Earthquakes: Radiated Energy and the Physics
of Faulting (eds R. Abercrombie, A. McGarr, G. Di Toro and H. Kanamori). doi:10.1029/170GM22
Simis, M. J., Madden, H., Cacciatore, M. A., & Yeo, S. K. (2016). The lure of rationality: Why does the deficit
model persist in science communication? Public Understanding of Science, 25(4), 400–414.
https://doi.org/10.1177/0963662516629749
Szolucha, A. (2019). "A social take on unconventional resources: Materiality, alienation and the making of
shale gas in Poland and the United Kingdom." Energy Research & Social Science 57: 101254.
Stephenson, M. H., P. Ringrose, S. Geiger, M. Bridden and D. Schofield (2019). "Geoscience and
decarbonization: current status and future directions." Petroleum Geoscience 25(4): 501.
Stewart, S., Allen, P. A 20-km-diameter multi-ringed impact structure in the North Sea. Nature 418, 520–
523 (2002). https://doi.org/10.1038/nature00914
Stewart, S., Allen, P. An alternative origin for the 'Silverpit crater' (reply). Nature 428, 2 (2004).
https://doi.org/10.1038/nature02480
Tang C.A., Kaiser P.K., 1998. Numerical simulation of cumulative damage and seismic energy release
during brittle rock failure—Part I: fundamentals, International Journal of Rock Mechanics and Mining
Science, 35, 113–121.
Taylor, H. A., Renshaw, C. E., & Jensen, M. D. (1997). Effects of computer-based role-playing on decision
making skills. Journal of Educational Computing Research, 17, 147 - 164.
Taylor, A. L., S. Dessai and W. Bruine de Bruin (2014). "Public perception of climate risk and adaptation in
the UK: A review of the literature." Climate Risk Management 4-5: 1-16.
TFSG (2015) Task Force on Shale Gas [TFSG]. Assessing the Impact of Shale Gas on the Local Environment
and Health. Second Interim Report. London, UK.
Thomas, M., T. Partridge, B. H. Harthorn and N. Pidgeon (2017). "Deliberating the perceived risks, benefits,
and societal implications of shale gas and oil extraction by hydraulic fracturing in the US and UK." Nature
Energy 2: 17054.
Trutnevyte, E. & Ejderyan, O. (2018) Managing geoenergy-induced seismicity with society, Journal of Risk
Research, 21:10, 1287-1294, DOI: 10.1080/13669877.2017.1304979
Tversky, A. and D. Kahneman (1981). "The framing of decisions and the psychology of choice." Science
1008  211(4481): 453.

Van de Graaf, T., Haesebrouck, T., & Debaere, P. (2018) Fractured politics? The comparative regulation of
shale gas in Europe, Journal of European Public Policy, 25:9, 1276-1293, DOI:
1011  10.1080/13501763.2017.1301985

van Loon, A.J., 2000. The stolen sequence. Earth-Science Reviews, 52(1-3), pp.237-244.
Vander Beken, T., Dorn, N. and Van Daele, S., 2010. Security risks in nuclear waste management:
Exceptionalism, opaqueness and vulnerability. Journal of environmental management, 91(4), pp.940-948.





Venhuizen, G.J., Hut, R., al.bers, C., Stoof, C.R. and Smeets, I., 2019. Flooded by jargon: how the
interpretation of water-related terms differs between hydrology experts and the general
audience. Hydrology and Earth System Sciences, 23(1), pp.393-403.
Vergara, W., Rios, A.R., Paliza, L.M.G., Gutman, P., Isbell, P., Suding, P.H. and Samaniego, J., 2013. The
climate and development challenge for Latin America and the Caribbean: options for climate-resilient,
low-carbon development. Inter-American Development Bank.
Warpinski NR, Du J, Zimmer U (2012) Measurements of hydraulic-fracture-induced seismicity in gas shales.
SPE Prod Oper 27:240–252
Westaway, R. and P. L. Younger (2014). "Quantification of potential macroseismic effects of the induced
seismicity that might result from hydraulic fracturing for shale gas exploitation in the UK." Quarterly
Journal of Engineering Geology and Hydrogeology 47(4): 333-350.
Williams, L., P. Macnaghten, R. Davies and S. Curtis (2017). "Framing 'fracking': Exploring public
perceptions of hydraulic fracturing in the United Kingdom." Public Understanding of Science 26(1): 89-
1028   104.

Whitmarsh, L., N. Nash, P. Upham, A. Lloyd, J. P. Verdon and J. M. Kendall (2015). "UK public perceptions
of shale gas hydraulic fracturing: The role of audience, message and contextual factors on risk perceptions
and policy support." Applied Energy 160(Supplement C): 419-430.
Yeo, R., 2003. Defining science: William Whewell, natural knowledge and public debate in early Victorian
Britain (Vol. 27). Cambridge University Press.
Underhill, J. An alternative origin for the 'Silverpit crater'. Nature 428, 1–2 (2004).
https://doi.org/10.1038/nature02476