# Peer review of "Fracking bad language: Hydraulic fracturing and earthquake risks"

_Geoscience Communication, 2020_

## Short Comment (SC1) · 8 Sep 2020

The paper presents an interesting study into the language used to describe hydraulic fracturing-induced seismicity, and how that language, and the understanding of that language, differs between experts and the general public.

I feel that the provision of a little more context as to the history of hydraulic fracturing, and the concomitant history of hydraulic fracturing-induced seismicity (HF-IS hereafter), would benefit the paper. In particular, while the focus of this study is on the views of the UK public, a slightly more global view may still be required because, while the UK public will likely be impacted primarily by newsworthy events in the UK, most experts are likely to have followed the development of the industry across the world (especially

since the UK, with only 3 shale wells ever stimulated, represents a very small part of the world's shale gas story). To add some context, I provide some brief comments on the history of HF-IS below. I also recommend Verdon and Bommer (2020) for further details of cases of HF-IS around the world.

HF-IS is dependent on certain geological/geomechanical conditions being met: (i) pre-existing tectonic faults must be present in, or near to, the target reservoir; (ii) the stress conditions in the reservoir (or surrounding formations) must be conducive to generating slip (having a high shear stress relative to normal stress); (iii) the frictional properties of the faults must be such that any slip is accommodated by rapid rupture and the release of seismic energy; (iv) the perturbation generated by the hydraulic fracturing operations must be sufficient to induce slip on nearby faults.

Hence, the likelihood of generating HF-IS will be strongly dependent on the particular geological conditions within a specific reservoir. It will also depend on the nature of the specific hydraulic fracturing operation – "hydraulic fracturing" and "fracking" are commonly-used catch-all terms, but in practice may describe operations ranging from the use of a few 100 m3 via a vertical well in a "conventional" reservoir, to the use of tens of thousands of m3 via multi-stage stimulation in horizontal wells targeting a shale reservoir (with many orders of magnitude difference in permeability between "conventional" and "shale" formations).

As a result, the occurrence of HF-IS is observed to vary significantly. Many major shale plays, such as the Barnett (Texas), Bakken (North Dakota), and Marcellus (Pennsylvannia), have experienced little to no HF-IS despite thousands of wells being drilled and hydraulically stimulated (Verdon et al., 2016; van der Baan and Calixto, 2017; Skoumal et al., 2018). In contrast, hydraulic fracturing of the Duvernay and Montney formations has generated HF-IS with magnitudes ranging between 4 < M < 5 (e.g., Kao et al., 2018). In the Sichuan Basin, China, hydraulic fracturing has generated HF-IS with M > 5, although debate continues as to whether events are actually triggered by hydraulic fracturing or salt dissolution mining (e.g., Lei et al., 2019). It is certainly not true to

claim, as the authors do on line 183, that the UK has experienced the highest recorded magnitudes of HF-IS.

Rates of HF-IS are even observed to vary significantly even within a basin – for example, while stimulation of the Montney and Duvernay shales in the West Canadian Sedimentary Basin (WCSB) has generated numerous cases of induced seismicity, extensive hydraulic fracturing has also taken place overlying formations such as the Cardium and Mannville (tight gas sandstones) without generating any recorded cases of HF-IS. Similarly, Skoumal et al. (2018) show a lack of HF-IS during stimulation of the Marcellus formation in the Appalachian Basin, but that stimulation of the underlying Utica shale has produced several cases.

Hydraulic fracturing has been used since the late 1940s (Montgomery and Smith, 2010). For much of this time, it was used in conventional formations, and no cases of HF-IS were reported. During the 2000s, hydraulic fracturing was adapted for use in shale formations. Among the most significant plays to be developed at the start of the shale gas "boom" was the Barnett, followed by the Marcellus and the Bakken. All three of these plays have been, in essence, aseismic. This, combined with the years of observations of aseismic hydraulic fracturing from conventional and tight gas formations, led the US National Research Council (2013) to state that "shale gas recovery does not pose a high risk for inducing felt seismic events (M > 2)".

The first case of HF-IS to be felt and widely reported were the events that occurred during hydraulic fracturing of the Preese Hall well in Lancashire in 2011 (Clarke et al., 2014). In fact, during this time cases of HF-IS were also occurring in the Horn River Basin, British Colombia (e.g., Farahbod et al., 2015), although the Horn River events were not widely reported at the time (perhaps related to the fact that population density in north-eastern B.C. is extremely low). Since then, as the use of hydraulic fracturing began to be used in more shale gas plays around the world, more cases of HF-IS have been reported. All of the cases identified in Verdon and Bommer (2020) were induced by hydraulic fracturing in shale reservoirs, and we are not aware of any cases of felt

seismicity induced by hydraulic fracturing in tight sandstone or conventional reservoirs. This may be a factor of the lower volumes of fluid typically used for hydraulic fracturing in tight and conventional reservoirs, and the fact that HF-IS is typically observed to scale with injected volume (e.g., Schultz et al., 2018; Clarke et al., 2019).

Considering the cases identified in Table 1 of Verdon and Bommer (2020), most of these cases occurred (or at least were described in publications) from 2014 onwards. Hence, while I obviously can't speak on behalf of the US National Research Council, I very much doubt that, if asked to re-assess the risks if HF-IS today, that they would come to the same conclusion as they did in 2012. Given the timelines described above, the fact that the data collection for this study took place in 2014 makes it particularly interesting (or challenging, depending on one's perspective), since this would represent a time of flux in terms of our understanding of HF-IS. Given the conclusions of the US National Research Council (2013) study, it would not be unreasonable to expect experts to surmise that the risks of HF-IS were low. Eight years down the line from the US National Research Council study, our knowledge of the factors that influence HF-IS has grown substantially. For my own part, the question "do you associate shale gas with earthquakes?" would be met with the answer "that depends, both on the geomechanical characteristics of the formation being targeted, and the nature of the hydraulic fracturing operation being proposed" (which, as described above, can vary by orders of magnitude within the catch-all term "hydraulic fracturing").

Similarly, since 2014, attempts have also been made to harmonize the language used to describe seismic events of different magnitudes that might occur at shale gas sites (e.g., Eaton, 2018). It would be fascinating to know whether the expert judgements and public views might have shifted since 2014, though I presume a repeat of the work presented would now be difficult to do. However, I do feel that a little more context in terms of the state of the science on HF-IS, and where it was in 2014 compared to today, would help provide important context for the results.

References

Clarke, H., L. Eisner, P. Styles, P. Turner, 2014. Felt seismicity associated with shale gas hydraulic fracturing: The first documented example in Europe: Geophysical Research Letters 41, 8308-8314.

Clarke, H., J.P. Verdon, T. Kettlety, A.F. Baird, J-M. Kendall, 2019. Real time imaging, forecasting and management of human-induced seismicity at Preston New Road, Lancashire, England: Seismological Research Letters 90, 1902-1915.

Eaton, D.W., 2018. Passive seismic monitoring of induced seismicity: Cambridge University Press.

Farahbod, A.M., H. Kao, D.M. Walker, J.F. Cassidy, 2015. Investigation of regional seismicity before and after hydraulic fracturing in the Horn River Basin, northeast British Columbia: Canadian Journal of Earth Sciences 52: 112-122.

Kao, H., R. Visser, B. Smith, S. Venables, 2018. Performance assessment of the induced seismicity traffic light protocol for northeastern British Columbia and western Alberta: The Leading Edge 37, 117-126.

Lei, X., Z. Wang, J. Su, 2019b. Possible link between long-term and short-term water injections and earthquakes in salt mine and shale gas site in Changning, south Sichuan Basin, China: Earth and Planetary Physics 3, 510-525.

Montgomery, C.T. and M.B. Smith, 2010. Hydraulic fracturing history of an enduring technology: Journal of Petroleum Technology 62, 26-41.

National Research Council, 2013. Induced Seismicity Potential in Energy Technologies: The National Academies Press, Washington, DC.

Schultz, R., G. Atkinson, D.W. Eaton, Y.J. Gu, H. Kao, 2018. Hydraulic fracturing volume is associated with earthquake productivity in the Duvernay play: Science 359, 304-308.

Skoumal, R.J., M.R. Brudzinski, B.S. Currie, 2018a. Proximity of Precambrian base-

ment affects the likelihood of induced seismicity in the Appalachian, Illinois, and Williston basins, central and eastern United States: Geosphere 14, 1365-1379.

van der Baan, M. and F.J Calixto, 2017. Human-induced seismicity and large-scale hydrocarbon production in the USA and Canada: Geochemistry, Geophysics, Geosystems 18, 2467-2485.

Verdon, J.P. and J.J. Bommer, 2020. Green, yellow, red, or out of the blue? An assessment of Traffic Light Schemes to mitigate the impact of hydraulic fracturing-induced seismicity: Journal of Seismology, in press. Pre-print available at: https://www1.gly.bris.ac.uk/∼gljpv/PDFS/VerdonBommer_2020_JOSE.pdf

---

## Referee Comment (RC1) · Mark Ireland (Referee) · 9 Oct 2020

General comments The paper has a good structure and examines the topic of the language of induced seismicity in good detail. The paper should be of broad interest to Earth Scientists, and those in other disciplines. The article would benefit from perhaps a glossary of terms at the start, given the variety of terms used throughout. The results section which describes the survey results could possibly be more concise, given the tables and figures which also communicate the results. The main areas for possible improvement is clarity in the use of terms used when describing the shale gas activities (mentioned in the specific comments).

Specific Comments Lines 39/40 - in the introduction the authors introduce the stakeholders and include 'scientists. With specific reference to controversial geosciences, it may be useful to pick apart the different roles which scientist have in shale gas – for example, within industry, within academia, within the regulators.

The authors introduce that many geoscience concepts and technologies are unfamiliar to the public (line 49/50), but it may also be relevant to discuss here the contrast between established and 'new' activities. o this extent a discussion of changes in perceived acceptance – what may have been an acceptable in the past, is no longer socially perceived as acceptable (e.g. Beck et al. 1993),

Authors introduce disputes in geoscience, however, do not include here mention of the Lusi mud volcano (e.g. Tingay et al., 2018) – which is highly relevant given that it was a source of both geoscience, community and political contention.

The use of 'geological engineering' throughout may possibly lead to confusion, particularly given the broad appeal of the paper. It may make sense to use 'geologist' and 'engineer' separately, particularly in the case of hydraulic fracturing, where the two areas of expertise have different roles.

Line 82/83 - references 'the language in communicating shale gas extraction' – although this paper focuses on the language of surrounding induced seismicity, it seems likely that 'shale gas' more broadly is thwart with many examples of 'bad language'. For example, even the use of the word 'extraction' in the UK context and to hydraulic fracturing could result in confusion. The authors could expand on what they consider the term extraction to encompass. Does this include all elements of the E&P lifecycle?

The article should consider expanding the description of hydraulic fracturing, and consider describing the range of different techniques, e.g. King (2012). The article could also differentiate between hydrulic fracturing and other well stimulation techqniues. The addition of a diagram to illustrate the practice of hydraulic fracturing could also make the article more widely accessible. GCD
Since specific reference is made throughout to the induced seismicty in the UK, perhaps an examination of the language used in the Hydraulic Fracture Plans prepared by operators and provided to the OGA and EA could be included in the compilation of publicly available expert reports.

Line 145 - the term 'tight gas' is introduced and seems to be used to refer to shale gas. In the O&G industry, commonly the terms tight gas and shale gas are used to define different resources. tight gas commonly refers to a resevoir where the hydrocarbons are within a conventional scale pore space (e.g microns) but are not connected. Whereas in shale gas resources the pores are often nanometres scale, and, for example may include pore space within organic components of the shale,

Line 154/155 – "not all seismic events have any detectable effect in terms of being felt, or recorded" – this statement could be expanded to include references, and to mention what the detection limits are for seismic events.

Lines 156-167 – covers a discussion on quantifying seismicity. However, it would perhaps be appropriate here to discuss or make mention of other industries, such as quarrying, which have their limits set/defined by ground motion.

Lines 173/174 – should the 'UK network' be defined? Are you referring to the BGS seismometer network? What is the detection limit of the dedicated surface arrays installed at the shale gas sites?

Line 181/182 – Could you clarify if the induced seismicity is associated with HF or with the production, or both?

Line 182/183 – "However, the largest recorded induced seismic events associated with shale gas extraction activities" – as previous, it might be worthwhile clarifying earlier in the paper where hydraulic fracturing sits within the context of shale gas extraction activities.

Line 213/214 - the technical expertise listed again includes 'disciplines' that might
cause confusion. Geological Engineering – not a field or role common in O&G sector, Oil Field Services – would seem to be a catch all category, and could include petroleum engineer.

Line 384/385 – "since hydraulic fracturing, by definition, will induce (albeit small) seismic events, it could be argued that assertions such as "shale gas development is associated with earthquakes" are factual" - are all seismic events earthquakes? what is the definition of the earthquakes? a section addressing individual scientific questions/issues

Line 619 - 622 - perhaps it would be worthwhile providing definitions of these terms in a glossary of terms. Providing definitions of the terms you use.

Line 656 – "much more decided on the topic than the UK general public" – referring back to the statement in the introduction that experts have a greater appreciation of uncertainty, this is an interesting finding, perhaps warrants discussion.

Line 689 – It might be beneficial to introduce the concept of 'what constitutes an earthquake?' much earlier in the paper.

Technical Corrections Line 52 – 'such uncertainty' – previous sentence does not specifically which uncertainty you are referring to. Line 70 - typo 'we explore the perception of and terminology' Lines 84/85 – examples of other causes of induced seismicity need references. Line 133/134 - Should include reference for moratorium/ suspension on fracking. Lines 145 – examples of applications of hydraulic fracturing should include references. Are there examples of HF for water production? Line 148 – Davies & Cartwright, 2007 paper is not an appropriate reference here. Line 168/169 – perhaps it should be clarified 'hydraulic fracturing' is one step in the extraction process. HF doesn't result in extraction, that still requires a pressure drawdown to create a differential. Line 345 – missing close bracket - (micro-seismic events, seismicity, and earthquakes) Line 698/699 – as Fig 1, TLS is OGA not UK Government. Line 191 – should make it clear whether the '6 months following' is a 6 month moratorium, or 6 months GCD
after the induced seismicity. Figure 1 caption– in the figure caption, it states that the traffic light system is from UK Government. The TLS is from the Oil and Gas Authority (OGA) and the OGA is a government owned company Figure 2 caption – "…shale gas with earthquakes decreases, while the number of participants that…" should add in '2012-2014' to make it clear over what years.

---

## Referee Comment (RC2) · Anonymous Referee #2 · 15 Oct 2020

This is a well written manuscript that would be of interest to both natural and social scientists working on various aspects of fracking. It adds to the literature that attempts to understand knowledge of the pros of and cons of fracking. It does rely almost exclusively on UK data and scholarship which is a limitation of the paper. A substantial and growing literature on fracking in the US and to a lesser extent Europe exists and this should be better represented.

Specific concerns: 1) In section 2 on page 6, the authors discuss their sources for expert views of induced seismicity from fracking. They note in the second paragraph, "We do not consider peer-reviewed publications in scientific journals, since relevant outcomes should be captured within the expert reports." Then later on the page they state, "Most expert reports conclude that the risks of induced seismicity from fracking in

the UK are very low. It is therefore fair to conclude that there is scientific consensus that the risks of induced seismicity are low, lower or no different to other human-induced seismicity..." This seems problematic to me. To conclude that there is scientific consensus on a topic, without consulting the peer-reviewed academic literature does not make sense. While some of the reports will undoubtedly have some scientific information in them, there is also the potential for bias in those reports who are going to often be more sympathetic to industry positions. Academics often have different opinions than industry and government people, which they derive primarily from peer-reviewed journal articles. The authors themselves note this on page 20 (albeit in another context), "It would be fair to presume that most academics would source their information from research papers..." This lack of the use of peer-reviewed science gage the "expert" opinion on induced seismicity is a serious weakness of this study.

2) On page 9 the authors discuss language usage in survey questions and how that may affect how respondents answer the questions (e.g. the questions are emotionally phrased, leading, etc.). At the bottom of page 9, the authors note that term "earthquakes" "evoke imagery of destruction and disaster, whereas phrases like 'seismic activity'....are less threatening." This is, of course, true. However, the authors do not discuss that researchers may chose to use the word "earthquakes" rather than "seismic activity" or "induced seismicity" because not all members of the lay public will know what those phrases mean. This is a common issue in survey question construction and should be acknowledged. This is probably one of the reasons why you find that, on page 25, "Academics use the phrase earthquake far more than those employed in other sectors..."

3) In the discussion of the participants in section 3.1.1, it would be helpful if the authors could provide information on how many of the 387 participants were employed in industry, government, academia and so on.

---

## Referee Comment (RC3) · Anonymous Referee #3 · 17 Oct 2020

General Comments This is a clean, well-written paper and was a pleasure to read. The paper presents information that is relevant to the governance of hydraulic fracturing and other endeavors with negative (real or perceived) externalities. I think it will be of interest to the journal's audience. Care was taken in the data analysis and presentation and the abstract and title are accurate. I have issues with the framing of the paper and the interpretation of the data and will direct my comments to these areas. I am suggesting major revisions because I feel the questions presented here are serious, but addressing them may not take that long. I do like the paper and the subject matter. I hope my comments reflect my interest in it.

The paper posits that a shared language about seismicity would facilitate risk communication. In so doing, it recasts the venerable "knowledge deficit" model of science

communication into a concern about how the absence of a shared language can make science communication difficult. This despite the fact that the authors cite a paper about why the model persists and how to overcome it (Simis et al. 2016). Developing a shared language is not a bad aim in itself and I agree that their point about the messiness of language, but I think it is unlikely to yield the results that the authors desire. While I agree that consistent use of terminology is beneficial between peers, the feeling I take aware from the paper is that the authors do not consider the public to be peers. And they are not, in the professional sense; but members of the public are peers in the stakeholder sense. Questions of who would develop the shared language, define the terms, etc. loom large in the paper. I get the sense, based on comments about the "nuanced" understanding of experts compared to the public throughout the paper, that this would be a top-down exercise. This would replicate the knowledge deficit model in linguistic form. To be fair to the authors, they did not specify who should develop the language. I am reading between the lines on this point. The paper would be stronger, and my concerns allayed somewhat, if they outlined a procedure for how developing a shared languages should or could happened.

Regardless, the emphasis on developing a shared language ignores how political (and industrial) affiliations and values influence perceptions of risk and the assessment of scientific information. Indeed, the authors bemoan the fact that language is "susceptible to emotional loading and misinterpretation" (Lines 30-31). Unfortunately, the public, and experts, always interpret information through a field of values and personal consequences. There is a broad literature in this area of science communication. Dietz, McCright, and Dunlap are some names that spring to mind, but there are many other sources.

The above is a major concern for me in the paper. I also have concerns about how the authors used previous work to position their own and the analysis of the data. Please see below.

I am curious if the authors considered how politics and personal interests shaped responses to their surveys. I have witnessed industry scientists and industry-friendly government officials argue all the nuances of data in a bid to halt pending regulations, whereas people with different interests and values (non-industry affiliated academics and the public) argued for restrictions. This is common in US climate change and energy politics.

Politics seems an unavoidable factor in this type of research. Language is a not a neutral tool, but one that is used to achieve certain ends. I fear that faith in the rationality of language, and those who would use it, is misguided.

Specific Comments Lines 21-26 – Tom Dietz (and others) have discussed that information is understood through a filter of values. This section, and the paper, would be strengthened by considering that the public (indeed, the many publics) hold values that are different from industry scientists and thus interpret information about fracking and related issues differently.

Comparison of closed ended surveys and qualitative data. I find this section problematic in a few ways. The authors cast doubt on survey data by expressing concern about how the surveys were constructed and analyzed. However, they do not provide any evidence from survey methodology literature to support their claims. Otherwise, statements such as the following from lines 296-304 are unsupported: "results of these closed surveys should therefore be interpreted and compared with some caution."

Providing support for this skepticism is particularly important since the authors uncritically accept the results from qualitative research (at least here) and suggest that it provides a more accurate portrayal of public opinion. To support this, a more robust comparison and discussion, rooted in literature, of these methods is needed. (For full disclosure, I am primarily a qualitative researcher, so I tend to favor qualitative methods and I appreciate the authors' point that closed ended questions do not allow respondents to offer their full knowledge and experience about a subject.)

There are other issues to address in this section as well. The authors compare the

results of the surveys and the qualitative data, but these are apples and oranges measurements. They write on lines 330-332, "Deliberative and dialogic approaches find that concerns around the risk of induced seismicity are not as significant as the surveys suggest; while concerns around induced seismicity are raised, it is not a primary or dominant concern within the context of other perceived risks." Regarding the first part of this statement, there is no way to compare the level of concern in the surveys with the level of concern in the qualitative data. Each method uses different measures and the authors offer no way to compare them systematically. This is a major problem. The second part of the statement is also problematic in that, in at least one of the surveys I reviewed (Whitmarsh et al. 2015), there was no claim that induced seismicity is the public's major concern about fracking. Indeed, in the Whitmarsh et al. 2015 paper, respondents, as the authors mention (Line 289), found that on average, rated water contamination as more pressing concern than earthquakes (3.53 for water contamination versus 3.27 earthquakes on a 5-point scale, Table 2). However, this difference does not appear to be large and it would seem inaccurate to imply, as I feel that the authors have done here by not providing the measurements in the text, that the public is not nearly concerned about earthquakes as water contamination.

I understand that the authors are trying to carve out a spot for their own mixed methods research with this review. However, I recommend that they revisit this section and recast their claims, using methods literature as support. This section, as currently written, gives the impression that the authors have a bias for qualitative methodologies and perhaps even for the outcomes they perceive in the cited studies. I want to be clear that I am not suggesting this is actually the case; rather, I wonder if it is an artifact of their analytic approach, which I do think could be improved. I did think that lines 395-407 gave a more nuanced discussion of the surveys compared to the qualitative data.

Line 399 – The authors write, "In contrast [compared to expert assessments], evidence on the perceived risk of induced seismicity amongst lay publics is mixed." I do not

think this is true. Every piece of research the authors introduced notes that the public perceives risk related to fracking. Perhaps if the authors change the sentence to read something like, "Evidence on the amount (or level) of perceived risk...) But again, I don't see enough here to make comparisons of levels of risk perception between studies.

Line 476 – The authors write, "The public cohort were not intended to represent the perspectives of the general public." But then in Line 482, they compare the results of the survey with the Nottingham YouGov, which is meant representative of the general public. Although the authors say that the public respondents in their sample were meant to represent those who take their information from media sources, this comparison still seems inappropriate to make since the public they sample are self-selected to be at the conferences and meetings where they were encountered. They are more highly engaged on the topic.

Line 513 – Could you say more about how experts' views are polarized here?

Line 623-624 – This section where the authors report that some people thought their questions were "leading" or that the term earthquake was "way too strong" hint at boundary keeping and political motivations. It would be interesting who in the sample said these things.

Line 648-651 – The authors write, "Nonetheless, our results do shed light on the ambiguity in the language around induced seismicity and the confusion that this can cause, the differences between publics and expert views on the matter (and difficulties in assessing expertise), and the limitations of using close surveys to elicit views on risk. The authors mentioned a variety of terms that respondents in different sectors tended to favor. However, I did not see where they demonstrated actual confusion. (If this is in the paper, then I apologize, but I have missed it.) Some of this language, when taken in combination with criticisms about terms being too strong or questions having a leading quality, might suggest that some respondents are using minimizing language. How

much of the choice in terminology is a struggle for accuracy and how much is a struggle to frame the issue in a particular light? The paper would benefit from considering such questions.

Line 665-666 – The authors write that there is no consensus amongst their survey respondents about whether or not earthquakes are associated with shale gas. It would be interesting to know who the authors define consensus.

Line 722-724 – The statement about doubt over public concern does not follow from experts' nuanced understanding of risk. The authors should identify who used the surveys to imply that concern among the public is high. Who is making the claim? The researchers or other parties? "However, by examining the reasoning provided by participants to explain their responses, we find that in reality this is much more nuanced amongst experts, and thus public concern about risks of induced seismicity may not be as high as the results of previous surveys have been used to imply."

Technical Corrections I cannot locate a Whitmarsh et al. 2014 citation in the references, probably a typo.

Line 678 – typo here "event with a cause in media reporting of an event without any there being a scientific explanation for a"

Lines 683-684 – plural/singular "In particular, those who 'do not' associate earthquakes and shale gas question the 684 definition of an earthquakes."

---

## Author Comment (AC3) · 21 Dec 2020

**Review Comment 1**

Dear Dr Mark Ireland (Reviewer 1),

Thank you for your detailed and constructive review. We are glad that you agree that the paper is of broad interest to Earth Scientists and beyond.

Your technical comments will tighten up how we describe shale gas activities in our revised manuscript thank you. In your general comments you suggest a glossary of terms; we address this suggestion alongside our responses to your specific comments in the attached document.

Best wishes,

Jen Roberts (Corresponding Author)

Response to Reviewer 1, Dr Mark Ireland

In the table below, our response (in blue text) can be found beneath each of your specific comments and technical corrections (in black text) which we have numbered for ease of any further discussion.

| Specific Comments | |
|---|---|
| 1 | Lines 39/40 - in the introduction the authors introduce the stakeholders and include 'scientists'. With specific reference to controversial geosciences, it may be useful to pick apart the different roles which scientists have in shale gas – for example, within industry, within academia, within the regulators.

We will amend the text as you suggest. New text is "Effective dialogue between stakeholders, including academics, regulators, industry, policy makers and the publics, is crucial to tackle this challenge" |
| 2 | The authors introduce that many geoscience concepts and technologies are unfamiliar to the public (line 49/50), but it may also be relevant to discuss here the contrast between established and 'new' activities. To this extent a discussion of changes in perceived acceptance – what may have been acceptable in the past, is no longer socially perceived as acceptable (e.g. Beck et al. 1993)

In Lines 49/50 we are specifically referring to geological concepts. However, we agree that (evolving) technologies and applications are also relevant to include in the Introduction. Rather than refer to 'new' technologies, we prefer to refer 'unfamiliar' technologies (since hydraulic fracturing approaches have been used for decades, c.f. James Verdon's Short Comment on this paper). Further, in this paper we are focussed on perceived risk – and not acceptable risk. However, we agree with the general point and in the Discussion section for our revised manuscript we will include discussion of how perceived risk can evolve through time, and can be influenced through factors such as those raised by Reviewer 3 (politics, motivation). |
| 3 | Authors introduce disputes in geoscience, however, do not include here mention of the Lusi mud volcano (e.g. Tingay et al., 2018) – which is highly relevant given that it was a source of both geoscience, community and political contention. |

| | |
|---|---|
| | We agree, the Lusi mud volcano is a relevant case study and will add the suggested reference to the text into lines ~56. |
| 4 | The use of 'geological engineering' throughout may possibly lead to confusion, particularly given the broad appeal of the paper. It may make sense to use 'geologist' and 'engineer' separately, particularly in the case of hydraulic fracturing, where the two areas of expertise have different roles. |
| | We will go through the text and separate into these disciplines where appropriate. However, geological engineering is a commonly used term that includes all aspects of subsurface engineering including those outwith hydrocarbon and production. |
| 5 | Line 82/83 - references 'the language in communicating shale gas extraction' – although this paper focuses on the language surrounding induced seismicity, it seems likely that 'shale gas' more broadly is thwart with many examples of 'bad language'. For example, even the use of the word 'extraction' in the UK context and to hydraulic fracturing could result in confusion. The authors could expand on what they consider the term extraction to encompass. Does this include all elements of the E&P lifecycle? |
| | Given that we are writing for an audience that included non-geoscientists we were trying to avoid industry-specific terms or jargon. We had implicitly included all E&P into 'extraction'. However in the UK, the focus was on shale gas exploration rather than extraction. In lines 82/83 we will change this to "exploration and development", and we will double-check the specific language used throughout the manuscript. |
| 6 | The article should consider expanding the description of hydraulic fracturing, and consider describing the range of different techniques, e.g. King (2012). The article could also differentiate between hydraulic fracturing and other well stimulation techniques. The addition of a diagram to illustrate the practice of hydraulic fracturing could also make the article more widely accessible. |
| | In Section 1.2 we will include further detail on the hydraulic fracturing process and history of in the UK, as also suggested by Dr James Verdon in the Short Comment. |
| 7 | Since specific reference is made throughout to induced seismicty in the UK, perhaps an examination of the language used in the Hydraulic Fracture Plans prepared by operators and provided to the OGA and EA could be included in the compilation of publicly available expert reports. |
| | We considered this too, in our original research. However, we opted not to examine the language within the HFPs in our research because - although publicly available - HFPs are not public-facing expert-led reports intended to conclude or advise on the risk of seismicity - they are a permitting requirement that lays out the anticipated seismicity and how it will be managed. |
| 8 | Line 145 - the term 'tight gas' is introduced and seems to be used to refer to shale gas. In the O&G industry, commonly the terms tight gas and shale gas are used to define different resources. tight gas commonly refers to a resevoir where the hydrocarbons are within a conventional scale pore space (e.g microns) but are not connected. Whereas in shale gas resources the pores are often nanometres scale, and, for example may include pore space within organic components of the shale, |

| | |
|---|---|
| | *We will still give the example of 'tight gas' but include also specific reference to shale gas, so that the two are not confounded.* |
| 9 | Line 154/155 – "not all seismic events have any detectable effect in terms of being felt, or recorded" – this statement could be expanded to include references, and to mention what the detection limits are for seismic events.

*We will add references to this sentence, but detection limits are not so simple, as the following paragraphs in the paper lay out.* |
| 10 | Lines 156-167 – covers a discussion on quantifying seismicity. However, it would perhaps be appropriate here to discuss or make mention of other industries, such as quarrying, which have their limits set/defined by ground motion.

*We refer to other industries in the previous paragraph, see line 152 – 154, including citation of Westaway & Younger et al. (2014) which compare seismic limits for different industries.* |
| 11 | Lines 173/174 – should the 'UK network' be defined? Are you referring to the BGS seismometer network? What is the detection limit of the dedicated surface arrays installed at the shale gas sites?

*We were referring to the detection limit laid out in Kendall et al. (2019) and also the BGS website, which indeed refers to the BGS seismograph stations. We will specify this in the text. For the detection limit of the dedicated surface arrays, this depends on factors outlined in lines 179/180, and so, similar to our response to Comment 9 in this table, it's not so simple as to give a number here.* |
| 12 | Line 181/182 – Could you clarify if the induced seismicity is associated with HF or with the production, or both?

*This is an important distinction – we will clarify in the text.* |
| 13 | Line 182/183 – "However, the largest recorded induced seismic events associated with shale gas extraction activities" – as previous, it might be worthwhile clarifying earlier in the paper where hydraulic fracturing sits within the context of shale gas extraction activities.

*See response to comment 6. We will refer to the recently published paper by Verdon and Bommer (2020), which documents other occurrences of hydraulic fracturing induced seismicity.* |
| 14 | Line  213/214 – the technical expertise listed again includes 'disciplines' that might cause confusion. Geological Engineering – not a field or role common in O&G sector, Oil Field Services – would seem to be a catch all category, and could include petroleum engineer.

*See response to Comment 4 in this table. We are not sure how widespread knowledge of what 'oil field services' entails and so we will use this term but then specify geology, petroleum engineering, too.* |
| 15 | Line 384/385 – "since hydraulic fracturing, by definition, will induce (albeit small) seismic events, it could be argued that assertions such as "shale gas development is associated with earthquakes" are factual" - are all seismic events earthquakes? what |

| | is the definition of the earthquakes? a section addressing individual scientific questions/ issues |
|---|---|
| | *This sentence was also questioned by Dr James Verdon in his Short Comment. We will change the sentence to include caveat "depending on how 'earthquake' is defined".* |
| 16 | Line 619 – 622 – perhaps it would be worthwhile providing definitions of these terms in a glossary of terms. Providing definitions of the terms you use. |
| | *The problem here is that we cannot define the phrases that are used by the survey participants (who we are quoted in those lines). What is meant by the terms that they opt to use might differ from how we define them. It is therefore not appropriate to include these terms or our codes in a glossary.* |
| | *Clarifications to the text terminology that you (and the other reviewers) have suggested in your specific and technical comments, together with further detail on the HF process will tighten the language, thus removing the need for a glossary.* |
| 17 | Line 656 – "much more decided on the topic than the UK general public" – referring back to the statement in the introduction that experts have a greater appreciation of uncertainty, this is an interesting finding, perhaps warrants discussion. |
| | *We agree and will expand on this in the discussion, as you suggest.* |
| 18 | Line 689 – It might be beneficial to introduce the concept of 'what constitutes an earthquake?' much earlier in the paper. |
| | *In Section 1.2 we already introduce that a range of terms are used to describe seismicity, framed by the title of the Kendall et al (2019) paper 'how big is a small earthquake?'. We will raise the question 'what constitutes an earthquake?' more explicitly there.* |

**Technical Corrections**

| 1 | Line 52 – 'such uncertainty' – previous sentence does not specifically which uncertainty you are referring to. |
|---|---|
| | *We will change this to make more specific ('uncertainty due to geological heterogeneity').* |
| 2 | Line 70 - typo 'we explore the perception of and terminology' |
| | *We will add oxford commas to make this sentence easier for the reader.* |
| 3 | Lines 84/85 – examples of other causes of induced seismicity need references. |
| | *We will add references (the same as those in Section 1.2).* |
| 4 | Line 133/134 - Should include reference for moratorium/ suspension on fracking. |
| | *We will add the BEIS reference which we use elsewhere in the article.* |
| 5 | Lines 145 – examples of applications of hydraulic fracturing should include references. Are there examples of HF for water production? |
| | *There are; we will add the references.* |

| 6 | Line 148 – Davies & Cartwright, 2007 paper is not an appropriate reference here. |
|---|---|
| | We meant to cite a different Cartwright paper, but instead have replaced with Engelder & Lacazette (1990). |
| 7 | Line 168/169 – perhaps it should be clarified 'hydraulic fracturing' is one step in the extraction process. HF doesn't result in extraction, that still requires a pressure drawdown to create a differential. |
| | True. We will swap the wording to say "hydraulic fracturing for shale gas exploration and development". (see also response to General Comment #5). |
| 8 | Line 345 – missing close bracket - (micro-seismic events, seismicity, and earthquakes) |
| | Thank you. This will be rectified. |
| 9 | Line 698/699 – as Fig 1, TLS is OGA not UK Government. |
| | Thank you. We will change this. |
| 10 | Line 191 – should make it clear whether the '6 months following' is a 6 month moratorium, or 6 months after the induced seismicity. |
| | The sentence was missing the word 'for' which will clarify this. |
| 11 | Figure 1 caption– in the figure caption, it states that the traffic light system is from UK Government. The TLS is from the Oil and Gas Authority (OGA) and the OGA is a government owned company |
| | We will simply say 'the UK's TLS' rather than the UK governments. |
| 12 | Figure 2 caption – ": : :shale gas with earthquakes decreases, while the number of participants that: : :" should add in '2012-2014' to make it clear over what years. |
| | We will add this clarification to Figure 2 caption |

---

## Author Comment (AC4) · 23 Dec 2020

Dear Dr James Verdon,

Thanks for your comment on our paper, and for bringing your new publication, Verdon and Bommer (2020) (and its laudable title!) to our attention. Thanks also for presenting such an overview of the landscape of hydraulic fracturing induced seismicity (HF-IS) and its history.

Your comments echo Reviewers 1&2 who wish to see further context to hydraulic fracturing and HF-IS in the UK and internationally in the introduction. In our revised manuscript we plan to present an overview of the changing global landscape – and will refer to your paper. We will clearly link our results and discussion to this context.

[Figure]

Thanks for pointing out our erroneous claim that the UK experienced the highest recorded magnitudes for HF-IS. This is now rectified. This was meant to refer to the Preese Hall events being the first case of HF-IS to be felt, but we see from your comment and paper that there have been other such events in British Colombia that were not so widely reported at the time.

Thanks also for alerting the Eaton (2018) paper to our attention as an example of an attempt to harmonize the language used to describe seismic events of different magnitudes.

Finally, we agree that there is scope for further research in this topic, in particular to see how views (and language) might have shifted since 2014. Given the issues we raise in our study, we would not favour a repeat of our survey research. However, the development of a shared language framework would require input from a range of stakeholders, and research accompanying the framework development would, we think, shed interesting light on how perceived risks have evolved.

Best wishes,

Dr Jen Roberts (lead author)

---

## Author Comment (AC5) · 23 Dec 2020

**Reviewer Comment 2**

Dear Reviewer 2,

Thank you for your constructive review. We are glad that you feel that the manuscript is well written and will be of interest to both natural and social scientists working on various aspects of fracking and related topics.

Regarding your General Comments, the research does indeed rely on UK data and narratives. The UK makes an interesting case study which we make clear in the abstract and throughout the paper (see lines 138-140 for example). We understand why you view this as a limitation, but expanding the research to include international data and scholarship is outside the scope of this research. We agree, however, that we should draw on the research and perspectives around perceived risk of induced seismicity from the US and elsewhere in the discussion, and we will implement this suggestion in the revised manuscript.

We address your specific suggestions alongside our responses to your specific comments below.

Best wishes,

Jen Roberts (Corresponding Author)
* * *
Below, our response to each of your specific comments can be found beneath, in blue text.

**Specific concerns:**

1) In section 2 on page 6, the authors discuss their sources for expert views of induced seismicity from fracking. They note in the second paragraph, "We do not consider peer-reviewed publications in scientific journals, since relevant outcomes should be captured within the expert reports." Then later on the page they state, "Most expert reports conclude that the risks of induced seismicity from fracking in the UK are very low. It is therefore fair to conclude that there is scientific consensus that the risks of induced seismicity are low, lower or no different to other human-induced seismicity..." This seems problematic to me. To conclude that there is scientific consensus on a topic, without consulting the peer-reviewed academic literature does not make sense. While some of the reports will undoubtedly have some scientific information in them, there is also the potential for bias in those reports who are going to often be more sympathetic to industry positions. Academics often have different opinions than industry and government people, which they derive primarily from peer-reviewed journal articles. The authors themselves note this on page 20 (albeit in another context), "It would be fair to presume that most academics would source their information from research papers..." This lack of the use of peer-reviewed science gage the "expert" opinion on induced seismicity is a serious weakness of this study.

We disagree. The reports that we include in our study are expert-led, policy and public facing (and therefore publicly accessible) reports which draw on the many hundreds of peer-review publications to inform the recommendations and/or conclusions. These reports are open access and were led by academics, or were academic-advised, as we note in the paper (learned societies, expert panels, scientific enquiries). Peer reviewed publications are not public facing, nor are they necessarily publicly accessible, and do not advise on the general risks related to the shale gas industry. Rather, peer-reviewed publications form a body of

evidence which is synthesised in the expert-led reports to inform expert advice. Our key interest in these reports is the language used to communicate risks of induced seismicity to a range of stakeholders.

Regarding consensus, we will check use of phrasing around consensus / expertise throughout the manuscript. Since not all shale gas experts are necessarily scientists, we will also check the wording throughout the manuscript and replace the use of the word 'scientific consensus' with more appropriate phrasing such as 'general agreement amongst expert bodies'.

2) On page 9 the authors discuss language usage in survey questions and how that may affect how respondents answer the questions (e.g. the questions are emotionally phrased, leading, etc.). At the bottom of page 9, the authors note that term "earthquakes" "evoke imagery of destruction and disaster, whereas phrases like 'seismic activity'....are less threatening." This is, of course, true. However, the authors do not discuss that researchers may chose to use the word "earthquakes" rather than "seismic activity" or "induced seismicity" because not all members of the lay public will know what those phrases mean. This is a common issue in survey question construction and should be acknowledged. This is probably one of the reasons why you find that, on page 25, "Academics use the phrase earthquake far more than those employed in other sectors..."

Thank you for highlighting that academics might choose to use the word 'earthquake' for ease of communication and understanding. This is a very good point and we will include this in the discussion on page 9 and in following Discussion.

3) In the discussion of the participants in section 3.1.1, it would be helpful if the authors could provide information on how many of the 387 participants were employed in industry, government, academia and so on.

We agree this will be helpful, and we will provide this information in the revised manuscript.

---

## Author Comment (AC6) · 23 Dec 2020

**Reviewer Comment 3**

Dear Reviewer 3,

Thank you for your constructive and supportive review. We are glad that you enjoyed reading the manuscript and that you like the research despite the issues that you raise.

We respond to your general and specific comments in the attached document.

Best wishes,

Jennifer Roberts (Corresponding Author)
* * *
In the table below, our response (in blue text) can be found beneath each of your general and specific comments (in black text) which we have numbered for ease of any further discussion.

| | General comments |
|---|---|
| 1 | The paper posits that a shared language about seismicity would facilitate risk communication. In so doing, it recasts the venerable "knowledge deficit" model of science communication into a concern about how the absence of a shared language can make science communication difficult. This despite the fact that the authors cite a paper about why the model persists and how to overcome it (Simis et al. 2016). Developing a shared language is not a bad aim in itself and I agree that their point about the messiness of language, but I think it is unlikely to yield the results that the authors desire. While I agree that consistent use of terminology is beneficial between peers, the feeling I take aware from the paper is that the authors do not consider the public to be peers. And they are not, in the professional sense; but members of the public are peers in the stakeholder sense. |
| | The initial prompt for this research was the tendency for the narrative around shale gas to dismiss or simplify public concerns around negative impacts, including induced seismicity, or to talk about the publics as a homogenous body. We were motivated to find out how shale gas experts answered the same questions being asked of the public, to test if expert views can be simplified much like the publics. What we found was that the questions about seismicity be answered differently by different people depending on what the word "earthquake" means to them. |
| | We were therefore disappointed to read that you feel that the paper appears to recast the information deficit model, and does not cast the publics as peers. We had taken particular care around this framing and the language we used, such as noting the expertise that publics bring (Lines 217/8), referring to differences between expert and lay perspectives as 'apparent' (Line 66), criticising 'technocracy' (Lines 106 – 115), and making clear that language challenges cause problems amongst experts, too (Lines 119-120). |
| | Importantly, we feel, is the emphasis in our research that the shared language is not to 'benefit' publics by improving their understanding (i.e. filling their 'knowledge deficit'). Rather, a common language framework is needed to a) help all stakeholders to communicate with each other and b) for perceived risks of a range of stakeholders to be better captured or understood. i.e. developing a shared language framework is not to facilitate one-way (expert to public) risk communication, but to support multi-way communication and understanding amongst all stakeholders, of which the publics are one/several. |
| | From your comments we deduce that resolving this broad issue is a case of revisiting the text with 'fresh eyes', adding qualifiers, and addressing the points raised in the specific comments. Rereading the introduction, we can see cause for your concerns. For example, the premise that we set in Line 64 is too simplistic, and in the text place more emphasis on the challenges in perception of the |

| | |
|---|---|
| | subsurface concepts, and do not present much detail on factors influencing risk perception or, say, values. We will make sure to include this, as well as to check that reference to stakeholders includes the publics. |
| 2 | Questions of who would develop the shared language, define the terms, etc. loom large in the paper. I get the sense, based on comments about the "nuanced" understanding of experts compared to the public throughout the paper, that this would be a top-down exercise. This would replicate the knowledge deficit model in linguistic form. To be fair to the authors, they did not specify who should develop the language. I am reading between the lines on this point. The paper would be stronger, and my concerns allayed somewhat, if they outlined a procedure for how developing a shared languages should or could happened.

In the paper, we do not propose how a language framework should be formed; that is not within the scope of our work. We feel this would be really interesting follow on research (see also our Response to Short Comment by Dr James Verdon).

In fact, we are not sure that a blanket language framework would be appropriate; it might be that a shared framework is 'drawn up' amongst stakeholders on a site by site or regional basis.

Either way, we agree that a top down approach would not be appropriate – any framework developed by a top down approach would not be 'shared'. Arguably there are several top down frameworks or classifications already in circulation (line 166, line 694) but – as we find - these are not widely used. While we are cautious to propose an approach in our manuscript, we will draw on these points to add a sentence or two on this in the Discussion. |
| 3 | Regardless, the emphasis on developing a shared language ignores how political (and industrial) affiliations and values influence perceptions of risk and the assessment of scientific information. Indeed, the authors bemoan the fact that language is "susceptible to emotional loading and misinterpretation" (Lines 30-31). Unfortunately, the public, and experts, always interpret information through a field of values and personal consequences. There is a broad literature in this area of science communication. Dietz, McCright, and Dunlap are some names that spring to mind, but there are many other sources.

We are aware of research around politics, motivation, and risk perception. The YouGov surveys which we compare our data against find an association between responses and political affiliation (alongside demographic factors and so on). We notice that we do not specifically refer to, say, values in our text on factors that affect how individuals perceive information (Lines 121 – 126) and framing around shale gas more specifically (Lines 127 - 134). We agree that this is an oversight and should be included in the Introduction and also the Discussion. Again, we'd like to emphasise that a shared language wouldn't resolve these challenges, or align different frames, but would support or facilitate multi-way communication amongst stakeholders. With our 'fresh eyes' we will make sure that this is articulated as such in the revised manuscript. |
| 4 | I am curious if the authors considered how politics and personal interests shaped re-sponses to their surveys. I have witnessed industry scientists and industry-friendly government officials argue all the nuances of data in a bid to halt pending regulations, whereas people with different interests and values (non-industry affiliated academics and the public) argued for restrictions. This is common in US climate change and energy politics.

This would be interesting research; in particular how these interests shape the language chosen to justify their response. However this is beyond the scope of our paper and our research data. |
| 5 | Politics seems an unavoidable factor in this type of research. Language is a not a neutral tool, but one that is used to achieve certain ends. I fear that faith in the rationality of language, and those who would use it, is misguided. |

This is an important message. We were careful to articulate that clarifying language would not, in itself, resolve communication challenges that we highlight in the paper, and we do not posit that a shared language would, for example, reduce perceived risks. Rather, a shared language would be one step to facilitate risk communication and, in doing so, help to clarify our understanding of how the risks of induced seismicity are assessed, perceived and understood. Your comments suggest that you feel this needs greater emphasis, and, similar to Comment #3 response, we will go through the text to make sure this is articulated as such in the revised manuscript.

| | **Specific Comments** |
|---|---|
| 1 | Lines 21-26 – Tom Dietz (and others) have discussed that information is understood through a filter of values. This section, and the paper, would be strengthened by considering that the public (indeed, the many publics) hold values that are different from industry scientists and thus interpret information about fracking and related issues differently.

We agree, this is important, and agree that it needs expanding on in the Introduction and reflect on in the Discussion (see response to General Comment #3). |
| 2 | Comparison of closed ended surveys and qualitative data. I find this section problematic in a few ways. The authors cast doubt on survey data by expressing concern about how the surveys were constructed and analyzed. However, they do not provide any evidence from survey methodology literature to support their claims. Otherwise, statements such as the following from lines 296-304 are unsupported: "results of these closed surveys should therefore be interpreted and compared with some caution."

While closed surveys must always be treated with care (and we will bring in methodology literature as you suggest), we present a clear case in the manuscript for why the results of closed surveys *that use ambiguous language* might be treated with extra care. We do not mean to imply that the authors of these surveys are not careful in how they interpret the data, nor how they executed their study; as Reviewer 2 points out the word 'earthquake' might have been chosen because it's a familiar, jargon-free, phrase (we will build this into the revised manuscript). However - as we show in our research - the term 'earthquake' means different things to different people, thus potentially muddying the understanding gained from any approach that uses such phrases without definition. |
| 3 | Providing support for this skepticism is particularly important since the authors uncritically accept the results from qualitative research (at least here) and suggest that it provides a more accurate portrayal of public opinion. To support this, a more robust comparison and discussion, rooted in literature, of these methods is needed. (For full disclosure, I am primarily a qualitative researcher, so I tend to favor qualitative methods and I appreciate the authors' point that closed ended questions do not allow respondents to offer their full knowledge and experience about a subject.)

You are correct that linking into research methods literature is called for in the discussion, and we will incorporate this into the revised manuscript. We are surprised that you feel we were uncritical towards qualitative research. In the manuscript, we certainly critique the reporting of qualitative research in terms of masking the phrasing used by participants to describe seismicity (although, to be fair, language wasn't the focus of their studies). Either way, regardless of the research approach, we are cautioning against the use of ambiguous language and terminology. Regardless of the research method used, questions about earthquakes will be answered differently depending on what the word means to the participant. |
| 4 | There are other issues to address in this section as well. The authors compare the results of the surveys and the qualitative data, but these are apples and oranges measurements. They write on lines 330-332, "Deliberative and dialogic approaches find that concerns around the risk of induced seismicity are not as significant as the surveys suggest; while concerns around induced seismicity are raised, it is not a primary or dominant concern within the context of other perceived risks." |

| | |
|---|---|
| | Regarding the first part of this statement, there is no way to compare the level of concern in the surveys with the level of concern in the qualitative data. Each method uses different measures and the authors offer no way to compare them systematically. This is a major problem.

This is an important point, and we will remedy the language and the message here so that we are not, as you put it, comparing apples and oranges with regards to the relative levels of concern. However we do not feel that these limitations restrict us from being able to synthesise broad themes from these different approaches and studies. |
| 5 | The second part of the statement is also problematic in that, in at least one of the surveys I reviewed (Whitmarsh et al. 2015), there was no claim that induced seismicity is the public's major concern about fracking. Indeed, in the Whitmarsh et al. 2015 paper, respondents, as the authors mention (Line 289), found that on average, rated water contamination as more pressing concern than earthquakes (3.53 for water contamination versus 3.27 earthquakes on a 5-point scale, Table 2). However, this difference does not appear to be large and it would seem inaccurate to imply, as I feel that the authors have done here by not providing the measurements in the text, that the public is not nearly concerned about earthquakes as water contamination.

You are correct that the reporting on the Whitmarsh et al., 2015 paper is slightly ambiguous and we will add in qualifiers about the relative scale of concern for earthquakes and water contamination. In the article, we do not claim that the surveys show that earthquakes *the* major concern, but "an important issue". It is difficult to say how important the issue is, when not all the issues of concern to publics are included in the survey. We will emphasise this in the revised manuscript, linking back to the methods literature. |
| 6 | I understand that the authors are trying to carve out a spot for their own mixed methods research with this review. However, I recommend that they revisit this section and recast their claims, using methods literature as support. This section, as currently written, gives the impression that the authors have a bias for qualitative methodologies and perhaps even for the outcomes they perceive in the cited studies. I want to be clear that I am not suggesting this is actually the case; rather, I wonder if it is an artifact of their analytic approach, which I do think could be improved. I did think that lines 395-407 gave a more nuanced discussion of the surveys compared to the qualitative data.

We are not trying to carve out a spot for our mixed methods research. We are trying to establish - from the literature and through survey - the perceived risk of seismicity from hydraulic fracturing and how this varies between stakeholders. We find that our understanding of perceived risks gets muddled by ambiguity around the language commonly used to describe seismicity.

We agree that a brief critique or overview of the strengths and weaknesses of qualitative approaches and closed question surveys would be appropriate to include in Section 2.1. We will also remove elements of analysis and discussion in Section 2.1 to make the approach clearer. |
| 7 | Line 399 – The authors write, "In contrast [compared to expert assessments], evidence on the perceived risk of induced seismicity amongst lay publics is mixed." I do notthink this is true. Every piece of research the authors introduced notes that the public perceives risk related to fracking. Perhaps if the authors change the sentence to read something like, "Evidence on the amount (or level) of perceived risk: : :) But again, I don't see enough here to make comparisons of levels of risk perception between studies.

This is a very fair point. All public perception studies report perceived risk of induced seismicity, and we will modify the text to reflect this. RE: comparing levels of perceived risks between studies, see response to Specific Comment #4 |
| 8 | Line 476 – The authors write, "The public cohort were not intended to represent the perspectives of the general public." But then in Line 482, they compare the results of the survey with the Nottingham YouGov, which is meant representative of the general public. Although the authors say |

that the public respondents in their sample were meant to represent those who take their information from media sources, this comparison still seems inappropriate to make since the public they sample are self-selected to be at the conferences and meetings where they were encountered. They are more highly engaged on the topic.

We make clear in the article that the 'lay public' in our sample are not representative of the general public (see Line 476). We compare all closed question responses (across all specialist conferences and public events) with the YouGov surveys to see whether and how participant views (our surveys) compare the general public (YouGov).

| 9 | Line 513 – Could you say more about how experts' views are polarized here? |
|---|---|
| | I suspect that the phrase "these experts" has led to ambiguity here. We will modify the sentence to clarify which experts we are referring to. In the preceding sentence, we detail how experts who obtain their information from research papers answer the closed question: 49% do; 47% do not (shown in Figure 3C). A very small proportion (4%) of this group are undecided. Thus, it might be perceived that these experts have split views. However, as we explore in the next subsection (3.2.2), the qualitative responses suggest that this apparent polarization is an artefact of language ambiguity. As such, we will add the qualifier that the experts' views are *apparently* polarized. |
| 10 | Line 623-624 – This section where the authors report that some people thought their questions were "leading" or that the term earthquake was "way too strong" hint at boundary keeping and political motivations. It would be interesting who in the sample said these things. |
| | Yes, interesting! However it won't be possible to reputably infer this from our data since we gathered no information about, for example, political motivation. Further, the question that they were asked is technically leading in how it was phrased and the issue of magnitude (and thus whether the word earthquake is appropriate or 'way too strong') is a technicality, too. |
| 11 | Line 648-651 – The authors write, "Nonetheless, our results do shed light on the ambiguity in the language around induced seismicity and the confusion that this can cause, the differences between publics and expert views on the matter (and difficulties in assessing expertise), and the limitations of using close surveys to elicit views on risk". The authors mentioned a variety of terms that respondents in different sectors tended to favor. However, I did not see where they demonstrated actual confusion. (If this is in the paper, then I apologize, but I have missed it.) Some of this language, when taken in combination with criticisms about terms being too strong or questions having a leading quality, might suggest that some respondents are using minimizing language. How much of the choice in terminology is a struggle for accuracy and how much is a struggle to frame the issue in a particular light? The paper would benefit from considering such questions. |
| | It is not the survey respondents who are confused. The survey results can be essentially flawed due to language ambiguity, which could lead to confusion on the perceived risks around induced seismicity. Take the example of two experts who perceive similar levels of risk around induced seismicity, but giving two different responses: one essentially says that they "do associate shale gas with earthquakes, but any earthquakes will be microseismic and will not be felt". The other says they "do not associate shale gas with earthquakes, any induced seismicity will be microseismic, and will not be felt".

The question 'how much of the choice in terminology is a struggle for accuracy and how much is a struggle to frame the issue in a particular light?' is important, and we agree that we should consider this in the paper. This links into political motivation and other such values previously raised. Using the example above, the two experts could have the same views about the risks of seismicity posed by hydraulic fracturing, but answer the yes/no question (do you associate shale gas with earthquakes?) differently because either (a) they have different preferred definitions for the term earthquake (b) they have different views about the shale gas industry, with one in favour and one opposed. |

| 12 | Line 665-666 – The authors write that there is no consensus amongst their survey respondents about whether or not earthquakes are associated with shale gas. It would be interesting to know who the authors define consensus. |
|---|---|
| | The use of the phrase 'consensus' was picked up by Reviewer 2 also. We will look to the peer reviewed literature for guidance on how 'consensus' and 'general agreement' are defined, and use an appropriately defined phrase in the manuscript. In any case, in the context of Line 165 any consensus would be 'apparent' given the issues we highlight in our paper. |
| 13 | Line 722-724 – The statement about doubt over public concern does not follow from experts' nuanced understanding of risk. The authors should identify who used the surveys to imply that concern among the public is high. Who is making the claim? The researchers or other parties? "However, by examining the reasoning provided by participants to explain their responses, we find that in reality this is much more nuanced amongst experts, and thus public concern about risks of induced seismicity may not be as high as the results of previous surveys have been used to imply." |
| | Firstly, our sentence shouldn't state "amongst experts"; our surveys of (informed) publics also show nuance. This is our mistake. We will add references to the studies that conclude (from their research) that publics are concerned about risk of earthquakes. We were not referring to our study here. |
| | **Technical comments** |
| 1 | I cannot locate a Whitmarsh et al. 2014 citation in the references, probably a typo. |
| | Whitmarsh et al (2014) is a technical report. Our apologies for leaving out of the reference list. We will double check whether the 2014 report or associated 2015 publication is most appropriate. |
| 2 | Line 678 – typo here "event with a cause in media reporting of an event without any there being a scientific explanation for a" |
| | It's not a typo, but a very confusing sentence. We will find a better phrasing. |
| 3 | Lines 683-684 – plural/singular "In particular, those who 'do not' associate earthquakes and shale gas question the 684 definition of an earthquakes." |
| | Thanks. Have replaced with "definition of an earthquake". |

---

## Author Response (AR1)

**Point by Point Response to Comments**

In the following, the 'revised manuscript' line numbers correspond to the line numbers in the revised manuscript with track changes (simply mark-up).

**Review Comment 1**

In the table below, we outline each of Reviewer 1 (R1) comments (in black text), and our response (in blue text) beneath.

| | |
|---|---|
| 1 | Lines 39/40 - in the introduction the authors introduce the stakeholders and include 'scientists'. With specific reference to controversial geosciences, it may be useful to pick apart the different roles which scientists have in shale gas – for example, within industry, within academia, within the regulators. |
| | We have amended the text as R1 suggests to become "Effective dialogue between stakeholders, including academics, regulators, industry, policy makers and the publics, is crucial to tackle this challenge" [revised manuscript lines 38] |
| 2 | The authors introduce that many geoscience concepts and technologies are unfamiliar to the public (line 49/50), but it may also be relevant to discuss here the contrast between established and 'new' activities. To this extent a discussion of changes in perceived acceptance – what may have been acceptable in the past, is no longer socially perceived as acceptable (e.g. Beck et al. 1993) |
| | In Lines 49/50 we are specifically referring to geological concepts. However, we agree that (evolving) technologies and applications are also relevant to include in the Introduction. Rather than refer to 'new' technologies, we prefer to refer 'unfamiliar' technologies (since hydraulic fracturing approaches have been used for decades, c.f. James Verdon's Short Comment on this paper). Further, in this paper we are focussed on perceived risk – and not acceptable risk. However, we agree with the general point and discuss temporal evolution of perceived risk with time in the revised manuscript discussion [revised manuscript lines 807] |
| 3 | Authors introduce disputes in geoscience, however, do not include here mention of the Lusi mud volcano (e.g. Tingay et al., 2018) – which is highly relevant given that it was a source of both geoscience, community and political contention. |
| | We agree, the Lusi mud volcano is a relevant case study and we have added the suggested reference to the text [revised manuscript lines 54] |
| 4 | The use of 'geological engineering' throughout may possibly lead to confusion, particularly given the broad appeal of the paper. It may make sense to use 'geologist' and 'engineer' separately, particularly in the case of hydraulic fracturing, where the two areas of expertise have different roles. |
| | We have gone through the text and separated into these disciplines where appropriate [e.g. see lines 224, 791 revised manuscript]. However, geological engineering is a commonly used term that includes all aspects of subsurface engineering including those outwith hydrocarbon and production. |
| 5 | Line 82/83 - references 'the language in communicating shale gas extraction' – although this paper focuses on the language surrounding induced seismicity, it |

seems likely that 'shale gas' more broadly is thwart with many examples of 'bad language'. For example, even the use of the word 'extraction' in the UK context and to hydraulic fracturing could result in confusion. The authors could expand on what they consider the term extraction to encompass. Does this include all elements of the E&P lifecycle?

Given that we are writing for an audience that included non-geoscientists we were trying to avoid industry-specific terms or jargon. We had implicitly included all E&P into 'extraction'. However in the UK, the focus was on shale gas exploration rather than extraction. In lines 82/83 we have changed this to "exploration and development", and we have double-checked the specific language used throughout the manuscript [for example, see revised manuscript line 77, 138]. For clarity we have also changed the language to consistently refer to boreholes (rather than wells).

| 6 | The article should consider expanding the description of hydraulic fracturing, and consider describing the range of different techniques, e.g. King (2012). The article could also differentiate between hydraulic fracturing and other well stimulation techniques. The addition of a diagram to illustrate the practice of hydraulic fracturing could also make the article more widely accessible. |
| | We have included further detail in Section 1.2 on the hydraulic fracturing process and history of in the UK, as also suggested by Dr James Verdon in the Short Comment [see revised manuscript lines 136] |
| 7 | Since specific reference is made throughout to induced seismicty in the UK, perhaps an examination of the language used in the Hydraulic Fracture Plans prepared by operators and provided to the OGA and EA could be included in the compilation of publicly available expert reports. |
| | We considered this too, in our original research. However, we opted not to examine the language within the HFPs in our research because - although publicly available - HFPs are not public-facing expert-led reports intended to conclude or advise on the risk of seismicity - they are a permitting requirement that lays out the anticipated seismicity and how it will be managed. |
| 8 | Line 145 - the term 'tight gas' is introduced and seems to be used to refer to shale gas. In the O&G industry, commonly the terms tight gas and shale gas are used to define different resources. tight gas commonly refers to a reservoir where the hydrocarbons are within a conventional scale pore space (e.g microns) but are not connected. Whereas in shale gas resources the pores are often nanometres scale, and, for example may include pore space within organic components of the shale, |
| | We have still used the example of 'tight gas' but have included also specific reference to shale gas, so that the two are not confounded [see revised manuscript lines 132]. |
| 9 | Line 154/155 – "not all seismic events have any detectable effect in terms of being felt, or recorded" – this statement could be expanded to include references, and to mention what the detection limits are for seismic events. |
| | Detection limits are not so simple, as the following paragraphs in the paper (with references) lay out. |

| | |
|---|---|
| 10 | Lines 156-167 – covers a discussion on quantifying seismicity. However, it would perhaps be appropriate here to discuss or make mention of other industries, such as quarrying, which have their limits set/defined by ground motion.

We refer to other industries in the previous paragraph, see line 152 – 154, including citation of Westaway & Younger et al. (2014) which compare seismic limits for different industries. |
| 11 | Lines 173/174 – should the 'UK network' be defined? Are you referring to the BGS seismometer network? What is the detection limit of the dedicated surface arrays installed at the shale gas sites?

We were referring to the detection limit laid out in Kendall et al. (2019) and also the BGS website, which indeed refers to the BGS seismograph stations. We will specify this in the text. For the detection limit of the dedicated surface arrays, this depends on factors outlined in lines 179/180, and so, similar to our response to Comment 9 in this table, it's not so simple as to give a number here. |
| 12 | Line 181/182 – Could you clarify if the induced seismicity is associated with HF or with the production, or both?

This is an important distinction – we have clarified in the revised text [revised manuscript lines 181] |
| 13 | Line 182/183 – "However, the largest recorded induced seismic events associated with shale gas extraction activities" – as previous, it might be worthwhile clarifying earlier in the paper where hydraulic fracturing sits within the context of shale gas extraction activities.

See response to Comment 6. We refer to the recently published paper by Verdon and Bommer (2020), which documents other occurrences of hydraulic fracturing induced seismicity, see revised manuscript lines 175. |
| 14 | Line  213/214 – the technical expertise listed again includes 'disciplines' that might cause confusion. Geological Engineering – not a field or role common in O&G sector, Oil Field Services – would seem to be a catch all category, and could include petroleum engineer.

See response to Comment 4 in this table. We are not sure how widespread knowledge of what 'oil field services' entails and so we have also specified geology, petroleum engineering, too, see revised manuscript lines 216. |
| 15 | Line 384/385 – "since hydraulic fracturing, by definition, will induce (albeit small) seismic events, it could be argued that assertions such as "shale gas development is associated with earthquakes" are factual" - are all seismic events earthquakes? what is the definition of the earthquakes? a section addressing individual scientific questions/ issues

This sentence was also questioned by Dr James Verdon in his Short Comment. We have changed the sentence to include caveat "depending on how 'earthquake' is defined", see revised manuscript line 395. |
| 16 | Line 619 – 622 – perhaps it would be worthwhile providing definitions of these terms in a glossary of terms. Providing definitions of the terms you use. |

| | |
|---|---|
| | The problem here is that we cannot define the phrases that are used by the survey participants (who we are quoted in those lines). What is meant by the terms that they opt to use might differ from how we define them. It is therefore not appropriate to include these terms or our codes in a glossary.

Clarifications to the text terminology that R1 (and the other reviewers) have suggested in your specific and technical comments, together with further detail on the HF process will tighten the language, thus removing the need for a glossary. |
| 17 | Line 656 – "much more decided on the topic than the UK general public" – referring back to the statement in the introduction that experts have a greater appreciation of uncertainty, this is an interesting finding, perhaps warrants discussion.

We agree and have expanded on this in the discussion [lines 688 in revised manuscript] |
| 18 | Line 689 – It might be beneficial to introduce the concept of 'what constitutes an earthquake?' much earlier in the paper.

In Section 1.2 we already introduce that a range of terms are used to describe seismicity, framed by the title of the Kendall et al (2019) paper 'how big is a small earthquake?'. We have raised the question 'what constitutes an earthquake?' more explicitly there [revised manuscript line 161] |

**Technical Corrections**

| | |
|---|---|
| 1 | Line 52 – 'such uncertainty' – previous sentence does not specifically which uncertainty you are referring to.

We have changed the wording to be more specific ('uncertainty due to geological heterogeneity'). |
| 2 | Line 70 - typo 'we explore the perception of and terminology'

We have added oxford commas to make this sentence easier for the reader. |
| 3 | Lines 84/85 – examples of other causes of induced seismicity need references.

We feel that the references are unnecessary; the relevance of these activities to induced seismicity are referred to in the references already cited in this sentence (Trutnevyte & Ejderyan, 2018; Stephenson et al., 2019) |
| 4 | Line 133/134 - Should include reference for moratorium/ suspension on fracking.

We have added the BEIS reference. |
| 5 | Lines 145 – examples of applications of hydraulic fracturing should include references. Are there examples of HF for water production?

There are; we have added the references [revised manuscript line 130] |
| 6 | Line 148 – Davies & Cartwright, 2007 paper is not an appropriate reference here.

We meant to cite a different Cartwright paper, but instead have replaced with Engelder & Lacazette (1990). |

| | |
|---|---|
| 7 | Line 168/169 – perhaps it should be clarified 'hydraulic fracturing' is one step in the extraction process. HF doesn't result in extraction, that still requires a pressure drawdown to create a differential.

True. We have clarified the wording (see also response to General Comment #5). |
| 8 | Line 345 – missing close bracket - (micro-seismic events, seismicity, and earthquakes)

This is now rectified. |
| 9 | Line 698/699 – as Fig 1, TLS is OGA not UK Government.

This is now rectified. |
| 10 | Line 191 – should make it clear whether the '6 months following' is a 6 month moratorium, or 6 months after the induced seismicity.

The sentence was missing the word 'for' (now rectified) which will clarify this point. |
| 11 | Figure 1 caption– in the figure caption, it states that the traffic light system is from UK Government. The TLS is from the Oil and Gas Authority (OGA) and the OGA is a government owned company

We have simply said 'the UK's TLS' rather than the UK governments. |
| 12 | Figure 2 caption – ": : :shale gas with earthquakes decreases, while the number of participants that: : :" should add in '2012-2014' to make it clear over what years.

We have added this clarification to Figure 2 caption. |

**Review Comment 2**

In the table below, we outline Reviewer 2 (R2) comments (in black text), and our response (in blue text) beneath.

| | |
|---|---|
| 1 | [The manuscript] does rely almost exclusively on UK data and scholarship which is a limitation of the paper. A substantial and growing literature on fracking in the US and to a lesser extent Europe exists and this should be better represented.

The research does indeed rely on UK data and narratives. The UK makes an interesting case study which we make clear in the abstract and throughout the paper (see lines 138-140 for example). We understand why R2 views this as a limitation, but expanding the research to include international data and scholarship is outside the scope of this research. Further, much of the international research has tended to look at public preferences or views rather than perceived risks, and rather than focus on risk of seismicity. We agree, however, that, where relevant, we should draw on the research and perspectives around perceived risk of induced seismicity from the US and elsewhere in the discussion. In the revised manuscript we compare international perspectives with regards to common preference influences (see lines 403+ of revised manuscript). |
| 2 | In section 2 on page 6, the authors discuss their sources for expert views of induced seismicity from fracking. They note in the second paragraph, "We do not consider peer-reviewed publications in scientific journals, since relevant outcomes should be captured within the expert reports." Then later on the page they state, "Most expert reports conclude that the risks of induced seismicity from fracking in the UK are very low. It is therefore fair to conclude that there is scientific consensus that the risks of induced seismicity are low, lower or no different to other human-induced seismicity..." This seems problematic to me. To conclude that there is scientific consensus on a topic, without consulting the peer-reviewed academic literature does not make sense. While some of the reports will undoubtedly have some scientific information in them, there is also the potential for bias in those reports who are going to often be more sympathetic to industry positions. Academics often have different opinions than industry and government people, which they derive primarily from peer-reviewed journal articles. The authors themselves note this on page 20 (albeit in another context), "It would be fair to presume that most academics would source their information from research papers..." This lack of the use of peer-reviewed science gage the "expert" opinion on induced seismicity is a serious weakness of this study.

We disagree. The reports that we include in our study are expert-led, policy and public facing (and therefore publicly accessible) reports which draw on the many hundreds of peer-review publications to inform the recommendations and/or conclusions. These reports are open access and were led by academics, or were academic-advised, as we note in the paper (learned societies, expert panels, scientific enquiries). Peer reviewed publications are not public facing, nor are they necessarily publicly accessible, and do not advise on the general risks related to the shale gas industry. Rather, peer-reviewed publications form a body of evidence which is synthesised in the expert-led reports to inform expert advice. Our key interest in these reports is the language used to communicate risks of induced seismicity to a range of stakeholders.

Regarding the word consensus, we have checked our phrasing around consensus / expertise throughout the manuscript, as well as 'scientists' (since not all shale gas |

| | |
|---|---|
| | experts are necessarily scientists). We have replaced phrases such as 'scientific consensus' with more appropriate phrasing such as 'general agreement amongst expert bodies'. See, for example, revised manuscript lines 256 and 407. |
| 3 | 2) On page 9 the authors discuss language usage in survey questions and how that may affect how respondents answer the questions (e.g. the questions are emotionally phrased, leading, etc.). At the bottom of page 9, the authors note that term "earthquakes" "evoke imagery of destruction and disaster, whereas phrases like 'seismic activity'....are less threatening." This is, of course, true. However, the authors do not discuss that researchers may chose to use the word "earthquakes" rather than "seismic activity" or "induced seismicity" because not all members of the lay public will know what those phrases mean. This is a common issue in survey question construction and should be acknowledged. This is probably one of the reasons why you find that, on page 25, "Academics use the phrase earthquake far more than those employed in other sectors..." |
| | R2 points out that the word 'earthquake' might be chosen in surveys for ease of communication and understanding. This is a very good point and we have raised this in Section 2.2 [revised manuscript lines 367] and in the Discussion [lines 745]. |
| 4 | In the discussion of the participants in section 3.1.1, it would be helpful if the authors could provide information on how many of the 387 participants were employed in industry, government, academia and so on. |
| | This information is already provided in the manuscript, Section 3.1.3 – Data Analysis. |

**Review Comment 3**

In the table below, we outline each of Reviewer 3 (R3) comments (in black text), and our response (in blue text) beneath.

| | **General comments** |
|---|---|
| 1 | The paper posits that a shared language about seismicity would facilitate risk communication. In so doing, it recasts the venerable "knowledge deficit" model of science communication into a concern about how the absence of a shared language can make science communication difficult. This despite the fact that the authors cite a paper about why the model persists and how to overcome it (Simis et al. 2016). Developing a shared language is not a bad aim in itself and I agree that their point about the messiness of language, but I think it is unlikely to yield the results that the authors desire. While I agree that consistent use of terminology is beneficial between peers, the feeling I take aware from the paper is that the authors do not consider the public to be peers. And they are not, in the professional sense; but members of the public are peers in the stakeholder sense. |
| | The initial prompt for this research was the tendency for the narrative around shale gas to dismiss or simplify public concerns around negative impacts, including induced seismicity, or to talk about the publics as a homogenous body. We were motivated to find out how shale gas experts answered the same questions being asked of the public, to test if expert views can be simplified much like the publics. What we found |

was that the questions about seismicity be answered differently by different people depending on what the word "earthquake" means to them.

We were therefore disappointed to read that R3 felt that the paper appears to recast the information deficit model, and does not cast the publics as peers. We had taken particular care around this framing and the language we used, such as noting the expertise that publics bring (Lines 217/8), referring to differences between expert and lay perspectives as 'apparent' (Line 66), criticising 'technocracy' (Lines 106 – 115), and making clear that language challenges cause problems amongst experts, too (Lines 119-120).

Importantly, we feel, is the emphasis in our research that the shared language is not to 'benefit' publics by improving their understanding (i.e. filling their 'knowledge deficit'). Rather, a common language framework is needed to a) help all stakeholders to communicate with each other and b) for perceived risks of a range of stakeholders to be better captured or understood. i.e. developing a shared language framework is not to facilitate one-way (expert to public) risk communication, but to support multi-way communication and understanding amongst all stakeholders, of which the publics are one/several.

From R3 comments we deduce that resolving this broad issue was a case of revisiting the text with 'fresh eyes', adding qualifiers, and addressing the points raised in R2's specific comments. Somewhat ironically, as both R1 and R3 point out, the language in our original manuscript needed tightening. As such, we have edited the text carefully, and have, for example, made clear near the start of the paper that reference to stakeholders includes the publics, and presented more detail throughout the paper about the factors that influence risk perception and values.

| 2 | Questions of who would develop the shared language, define the terms, etc. loom large in the paper. I get the sense, based on comments about the "nuanced" understanding of experts compared to the public throughout the paper, that this would be a top-down exercise. This would replicate the knowledge deficit model in linguistic form. To be fair to the authors, they did not specify who should develop the language. I am reading between the lines on this point. The paper would be stronger, and my concerns allayed somewhat, if they outlined a procedure for how developing a shared languages should or could happened.

In the paper, we do not propose how a language framework should be formed; that is not within the scope of our work. We feel this would be really interesting follow on research (see also our Response to Short Comment by Dr James Verdon).

In fact, a blanket language framework may not be appropriate; it might be that a shared framework is 'drawn up' amongst stakeholders on a site by site or regional basis. This is a question for further research to explore. Either way, we agree that a top down approach would not be appropriate – any framework developed by a top down approach would not be 'shared' (arguably there are several top down frameworks or classifications already in circulation as we refer to in our paper but – as we find - these are not widely used). This was already clear in the Discussion and see no reason to expand further. |
|---|---|
| 3 | Regardless, the emphasis on developing a shared language ignores how political (and industrial) affiliations and values influence perceptions of risk and the assessment of scientific information. Indeed, the authors bemoan the fact that language is "susceptible to emotional loading and misinterpretation" (Lines 30-31). |

| | |
|---|---|
| | Unfortunately, the public, and experts, always interpret information through a field of values and personal consequences. There is a broad literature in this area of science communication. Dietz, McCright, and Dunlap are some names that spring to mind, but there are many other sources. |
| | We are aware of research around politics, motivation, and risk perception. The YouGov surveys which we compare our data against find an association between responses and political affiliation (alongside demographic factors and so on). We notice that we do not specifically refer to, say, values in our text on factors that affect how individuals perceive information (Lines 121 – 126 original manuscript) and framing around shale gas more specifically (Lines 127 - 134 original manuscript). We agree that this is an oversight and this is now included in the Introduction and also the Discussion (see lines 114, 768 of revised manuscript). Again, we'd like to emphasise that a shared language wouldn't resolve these challenges, or align different frames, but would support or facilitate multi-way communication amongst stakeholders. We have checked that this is articulated as such in the revised manuscript. |
| 4 | I am curious if the authors considered how politics and personal interests shaped responses to their surveys. I have witnessed industry scientists and industry-friendly government officials argue all the nuances of data in a bid to halt pending regulations, whereas people with different interests and values (non-industry affiliated academics and the public) argued for restrictions. This is common in US climate change and energy politics. |
| | This would be interesting research; in particular how these interests shape the language chosen to justify their response. However, this is beyond the scope of our paper and our research data. |
| 5 | Politics seems an unavoidable factor in this type of research. Language is a not a neutral tool, but one that is used to achieve certain ends. I fear that faith in the rationality of language, and those who would use it, is misguided. |
| | This is an important message, and we have adapted R3's words into the revised articles, see lines 339 of revised manuscript. We were careful to articulate that clarifying language would not, in itself, resolve communication challenges that we highlight in the paper, and we do not posit that a shared language would, for example, reduce perceived risks. Rather, a shared language would be one step to facilitate risk communication and, in doing so, help to clarify our understanding of how the risks of induced seismicity are assessed, perceived and understood. R3's comments suggest that this needed greater emphasis, and, similar to Comment #3 response, we have gone through the text to make sure this is articulated as such in the revised manuscript. |
| | **Specific Comments** |
| 1 | Lines 21-26 – Tom Dietz (and others) have discussed that information is understood through a filter of values. This section, and the paper, would be strengthened by considering that the public (indeed, the many publics) hold values that are different from industry scientists and thus interpret information about fracking and related issues differently. |

| | |
|---|---|
| | We agree, this is important, and agree that it needs expanding on in the Introduction and reflect on in the Discussion. We have made these changes in the revised manuscript - see response to General Comment #3. |
| 2 | Comparison of closed ended surveys and qualitative data. I find this section problematic in a few ways. The authors cast doubt on survey data by expressing concern about how the surveys were constructed and analyzed. However, they do not provide any evidence from survey methodology literature to support their claims. Otherwise, statements such as the following from lines 296-304 are unsupported: "results of these closed surveys should therefore be interpreted and compared with some caution."

 While closed surveys must always be treated with care we present a clear case in the manuscript for why the results of closed surveys *that use ambiguous language* might be treated with extra care. We do not mean to imply that the authors of these surveys are not careful in how they interpret the data, nor how they executed their study; as R2 points out the word 'earthquake' might have been chosen because it's a familiar, jargon-free, phrase. However - as we show in our research - the term 'earthquake' means different things to different people, thus potentially muddying the understanding gained from any approach that uses such phrases without definition. We have pointed to methodology literature in the revised manuscript; see for example lines 303, 758 revised manuscript. |
| 3 | Providing support for this skepticism is particularly important since the authors uncritically accept the results from qualitative research (at least here) and suggest that it provides a more accurate portrayal of public opinion. To support this, a more robust comparison and discussion, rooted in literature, of these methods is needed. (For full disclosure, I am primarily a qualitative researcher, so I tend to favor qualitative methods and I appreciate the authors' point that closed ended questions do not allow respondents to offer their full knowledge and experience about a subject.)

 We are surprised that R3 felt we were uncritical towards qualitative research. In the manuscript, we certainly critique the *reporting* of qualitative research in terms of masking the phrasing used by participants to describe seismicity (although, to be fair, language wasn't the focus of their studies). Either way, regardless of the research approach, we are cautioning against the use of ambiguous language and terminology. Regardless of the research method used, questions about earthquakes will be answered differently depending on what the word means to the participant.

 In the revised manuscript we link to research methods literature in the Discussion of (see revised manuscript lines 756). |
| 4 | There are other issues to address in this section as well. The authors compare the results of the surveys and the qualitative data, but these are apples and oranges measurements. They write on lines 330-332, "Deliberative and dialogic approaches find that concerns around the risk of induced seismicity are not as significant as the surveys suggest; while concerns around induced seismicity are raised, it is not a primary or dominant concern within the context of other perceived risks." Regarding the first part of this statement, there is no way to compare the level of concern in the surveys with the level of concern in the qualitative data. Each method uses different measures and the authors offer no way to compare them systematically. This is a major problem. |

| | |
|---|---|
| | This is an important point, and we have rectified the language and the message here so that we are not, as R3 put it, comparing apples and oranges with regards to the relative levels of concern. However, we do not feel that these limitations restrict us from being able to synthesise broad themes from these different approaches and studies. |
| 5 | The second part of the statement is also problematic in that, in at least one of the surveys I reviewed (Whitmarsh et al. 2015), there was no claim that induced seismicity is the public's major concern about fracking. Indeed, in the Whitmarsh et al. 2015 paper, respondents, as the authors mention (Line 289), found that on average, rated water contamination as more pressing concern than earthquakes (3.53 for water contamination versus 3.27 earthquakes on a 5-point scale, Table 2). However, this difference does not appear to be large and it would seem inaccurate to imply, as I feel that the authors have done here by not providing the measurements in the text, that the public is not nearly concerned about earthquakes as water contamination. |
| | R3 is correct that the reporting on the Whitmarsh et al., 2015 paper is slightly ambiguous and we have added in qualifiers about the relative scale of concern for earthquakes and water contamination (revised manuscript lines 292). In the article, we do not claim that the surveys show that earthquakes *the* major concern, but "an important issue". It is difficult to say how important the issue is, when not all the issues of concern to publics are included in the survey. We have emphasised this in the revised manuscript. |
| 6 | I understand that the authors are trying to carve out a spot for their own mixed methods research with this review. However, I recommend that they revisit this section and recast their claims, using methods literature as support. This section, as currently written, gives the impression that the authors have a bias for qualitative methodologies and perhaps even for the outcomes they perceive in the cited studies. I want to be clear that I am not suggesting this is actually the case; rather, I wonder if it is an artifact of their analytic approach, which I do think could be improved. I did think that lines 395-407 gave a more nuanced discussion of the surveys compared to the qualitative data. |
| | We are not trying to carve out a spot for our mixed methods research. We are trying to establish - from the literature and through survey - the perceived risk of seismicity from hydraulic fracturing and how this varies between stakeholders. We find that our understanding of perceived risks gets muddled by ambiguity around the language commonly used to describe seismicity. |
| | We do not feel that a critique or summary of the strengths and weaknesses of qualitative approaches and closed question surveys is appropriate to include in Section 2.1; it will disrupt and distract from the article. However in the revised manuscript we do link to the methods literature, including common/well known limitations of closed survey approaches e.g. 301. We also remove elements of analysis and discussion in Section 2.1 to make the approach clearer. |
| 7 | Line 399 – The authors write, "In contrast [compared to expert assessments], evidence on the perceived risk of induced seismicity amongst lay publics is mixed." I do not think this is true. Every piece of research the authors introduced notes that the public perceives risk related to fracking. Perhaps if the authors change the sentence to read something like, "Evidence on the amount (or level) of perceived |

| | |
|---|---|
| | risk: : :) But again, I don't see enough here to make comparisons of levels of risk perception between studies.

R3 makes a very fair point. All public perception studies report perceived risk of induced seismicity, and we have modified the text to reflect this. e.g. see lines 246 of revised manuscript. RE: comparing levels of perceived risks between studies, see response to Specific Comment #4 |
| 8 | Line 476 – The authors write, "The public cohort were not intended to represent the perspectives of the general public." But then in Line 482, they compare the results of the survey with the Nottingham YouGov, which is meant representative of the general public. Although the authors say that the public respondents in their sample were meant to represent those who take their information from media sources, this comparison still seems inappropriate to make since the public they sample are self-selected to be at the conferences and meetings where they were encountered. They are more highly engaged on the topic.

We make clear in the article that the 'lay public' in our sample are not representative of the general public (see Line 476). We compare all closed question responses (across all specialist conferences and public events) with the YouGov surveys to see whether and how participant views (our surveys) compare the general public (YouGov). |
| 9 | Line 513 – Could you say more about how experts' views are polarized here?

I suspect that the phrase "these experts" has led to ambiguity here. We have modified the sentence to clarify which experts we are referring to. In the preceding sentence, we detail how experts who obtain their information from research papers answer the closed question: 49% do; 47% do not (shown in Figure 3C). A very small proportion (4%) of this group are undecided. Thus, it might be perceived that these experts have split views. However, as we explore in the next subsection (3.2.2), the qualitative responses suggest that this apparent polarization is an artefact of language ambiguity. As such, we have added the qualifier that the experts' views are *apparently* polarized (revised manuscript lines 527). |
| 10 | Line 623-624 – This section where the authors report that some people thought their questions were "leading" or that the term earthquake was "way too strong" hint at boundary keeping and political motivations. It would be interesting who in the sample said these things.

Yes, interesting! However, it won't be possible to reputably infer this from our data since we gathered no information about, for example, political motivation. Further, the question that they were asked is leading in how it was phrased and the issue of magnitude (and thus whether the word earthquake is appropriate or 'way too strong') is a technicality, too. |
| 11 | Line 648-651 – The authors write, "Nonetheless, our results do shed light on the ambiguity in the language around induced seismicity and the confusion that this can cause, the differences between publics and expert views on the matter (and difficulties in assessing expertise), and the limitations of using close surveys to elicit views on risk". The authors mentioned a variety of terms that respondents in different sectors tended to favor. However, I did not see where they demonstrated actual confusion. (If this is in the paper, then I apologize, but I have missed it.) Some of this language, when taken in combination with criticisms about terms being too |

strong or questions having a leading quality, might suggest that some respondents are using minimizing language. How much of the choice in terminology is a struggle for accuracy and how much is a struggle to frame the issue in a particular light? The paper would benefit from considering such questions.

It is not the survey respondents who are confused. The survey results can be essentially flawed due to language ambiguity, which could lead to confusion on the perceived risks around induced seismicity. Take the example of two experts who perceive similar levels of risk around induced seismicity, but giving two different responses: one essentially says that they "do associate shale gas with earthquakes, but any earthquakes will be microseismic and will not be felt". The other says they "do not associate shale gas with earthquakes, any induced seismicity will be microseismic, and will not be felt".

The question 'how much of the choice in terminology is a struggle for accuracy and how much is a struggle to frame the issue in a particular light?' is important, and links into political motivation and other such values previously raised. We have modified the revised manuscript to reflect this. Using the example we give above, the two experts could have the same views about the risks of seismicity posed by hydraulic fracturing, but answer the yes/no question (do you associate shale gas with earthquakes?) differently because either (a) they have different preferred definitions for the term earthquake (b) they have different views about the shale gas industry, with one in favour and one opposed.

In the revised manuscript, while we incorporate values and motivation in the discussion (revised manuscript lines 268) we also make explicit in the methods that we are not seeking to understand these aspects in the research (revised manuscript lines 231).

| 12 | Line 665-666 – The authors write that there is no consensus amongst their survey respondents about whether or not earthquakes are associated with shale gas. It would be interesting to know who the authors define consensus.

The use of the phrase 'consensus' was picked up by R2 also, and have modified the phrasing in the revised manuscript (opting instead for words such as broad agreement e.g. manuscript lines 428, 903). In any case, in the context of Line 165 any consensus would be 'apparent' given the issues we highlight in our paper. |
| --- | --- |
| 13 | Line 722-724 – The statement about doubt over public concern does not follow from experts' nuanced understanding of risk. The authors should identify who used the surveys to imply that concern among the public is high. Who is making the claim? The researchers or other parties? "However, by examining the reasoning provided by participants to explain their responses, we find that in reality this is much more nuanced amongst experts, and thus public concern about risks of induced seismicity may not be as high as the results of previous surveys have been used to imply."

Firstly, our sentence shouldn't state "amongst experts"; our surveys of (informed) publics also show nuance. This was our mistake. We have added references to the studies that conclude (from their research) that publics are concerned about risk of earthquakes. We were not referring to our study here.. |
| | **Technical comments** |
| 1 | I cannot locate a Whitmarsh et al. 2014 citation in the references, probably a typo. |

| | |
|---|---|
| | Whitmarsh et al (2014) is a technical report. Our apologies for leaving out of the reference list. It is now added. |
| 2 | Line 678 – typo here "event with a cause in media reporting of an event without any there being a scientific explanation for a"

It's not a typo, but a very confusing sentence. We have removed it from the revised manuscript. |
| 3 | Lines 683-684 – plural/singular "In particular, those who 'do not' associate earthquakes and shale gas question the 684 definition of an earthquakes."

Thanks. Have replaced with "definition of an earthquake". |

**Short Comment**

In the table below, we outline each of main points in James Verdon's (JV) Short Comment (in black text), and our response (in blue text) beneath.

| | |
|---|---|
| 1 | I feel that the provision of a little more context as to the history of hydraulic fracturing, and the concomitant history of hydraulic fracturing-induced seismicity (HF-IS hereafter), would benefit the paper. In particular, while the focus of this study is on the views of the UK public, a slightly more global view may still be required because, while the UK public will likely be impacted primarily by newsworthy events in the UK, most experts are likely to have followed the development of the industry across the world (especially since the UK, with only 3 shale wells ever stimulated, represents a very small part of the world's shale gas story).

 JV's comments echo two other reviewers who wish to see further context to hydraulic fracturing and HF-IS in the UK and internationally in the introduction. In our revised manuscript we plan to present an overview of the changing global landscape, and refer to JV's paper - see lines 175+ of revised manuscript. We then link our results and discussion (of the reports and the surveys) to this context, for example see lines 235 and 663 of revised manuscript. |
| 2 | It is not true to claim, as the authors do on line 183, that the UK has experienced the highest recorded magnitudes of HF-IS.

 Thanks JV for pointing out our erroneous claim that the UK experienced the highest recorded magnitudes for HF-IS. This is now rectified, see line 175 of revised manuscript. This was meant to refer to the Preese Hall events being the first case of HF-IS to be felt, but we see from JV's comment and paper that there might have been other such events in British Colombia that were not so widely reported at the time. |
| 3 | Since 2014, attempts have also been made to harmonize the language used to describe seismic events of different magnitudes that might occur at shale gas sites (e.g., Eaton, 2018).

 Thanks JV for bringing Eaton (2018) to our attention as an example of an attempt to harmonize the language used to describe seismic events of different magnitudes. Although we cannot access the book, we find Eaton et al (2016) paper which is specifically on induced seismicity very illuminating, and have included in the revised paper (lines 163, 728) |
| 4 | Considering the cases identified in Table 1 of Verdon and Bommer (2020), most of these cases occurred (or at least were described in publications) from 2014 onwards. Hence, while I obviously can't speak on behalf of the US National Research Council, I very much doubt that, if asked to re-assess the risks if HF-IS today, that they would come to the same conclusion as they did in 2012. Given the timelines described above, the fact that the data collection for this study took place in 2014 makes it particularly interesting (or challenging, depending on one's perspective), since this would represent a time of flux in terms of our understanding of HF-IS. Given the conclusions of the US National Research Council |

(2013) study, it would not be unreasonable to expect experts to surmise that the risks of HF-IS were low. Eight years down the line from the US National Research Council study, our knowledge of the factors that influence HF-IS has grown substantially. For my own part, the question "do you associate shale gas with earthquakes?" would be met with the answer "that depends, both on the geomechanical characteristics of the formation being targeted, and the nature of the hydraulic fracturing operation being proposed" (which, as described above, can vary by orders of magnitude within the catch-all term "hydraulic fracturing").

This is helpful, and we have now captured this state of 'flux' as JV put it, or knowledge evolution, in the revised manuscript, and this has shaped how we present the findings, too. We also capture the desire for the ability to select 'that depends' in the survey responses (see Discussion line 684 revised manuscript).

---

## Author Response (AR2)

**Revision for Editor**

In the table below, our response (in blue text) can be found beneath each of your specific comments (in black text) which we have numbered for ease of any further discussion. We also provide the track changes version of the article.

|  | **General comments** |
|---|---|
| 1 | Add commas after et al. in citations throughout the paper when followed by the year.

Done |
| 2 | Reduce the use of parenthetical and dashed phrases within sentences. I've highlighted cases where these are distracting and can be reworded or the () or -- substituted with commas.

Done |
| 3 | In lists, use and or or after the last comma.

Done |
| 4 | Take out unnecessary words or phrases to shorten sentences and improve readability

Done |
| 5 | Eliminate or replace ambiguous language

Done |
| 6 | Make and back up all points clearly in the text so that the tables serve as a supplement but are not required to understand your arguments while reading the body of the paper

Done |
| 7 | Clarify when you are talking about only respondents from the conferences vs. the outreach events

Done |
| 8 | Cf. refers to 'compare to'--I think what you mean here is 'see for example'; please change all instances to 'see for example' or 'e.g.,'.

Done |
| 9 | Use punctuation consistently in table, i.e., add periods to the end of all sentences

Done |
|  | **Specific comments** |

| | |
|---|---|
| 10 | 1 "risk of seismicity" itself is ambiguous. Does this mean potential for damage or casualties? Or is it, like the other language you address in the article, left ambiguous by stakeholders and researchers?

Good point. We have replaced with "The potential to induce seismicity". |
| 11 | 13 Add in what year "halted shale gas operations and triggered moratoria." If more than one year and specificity is difficult, change "halted" to "has halted".

Changed to 'has halted' as you suggest. |
| 12 | 13-16 Please clarify: is the disconnect between the level of risk vs. concern about that risk, or between levels of risk perceived by publics vs. expert groups? Do you mean to have the word "concern" in this sentence? This could read that there is a disconnect between the level of concern *perceived* by publics and reported (about whom?) by expert groups. This sentence would benefit from removing unnecessary words, using more precise language if possible, and potentially breaking it up into two or more sentences.

We agree, this could be worded more clearly. At this point, we don't know if the disconnect is due to differences in the perceived risk, or the acceptability of that risk (concern). Taking on board your suggestions, we have replaced with:

"Prior to 2018 there seemed to be a disconnect between the conclusions of expert groups about the risk of adverse impacts from hydraulic fracturing induced seismicity, and the reported level of public concern about hydraulic fracturing induced seismicity. Further, a range of terminology was used to describe the induced seismicity (including tremors, earthquakes, seismic events, and micro-earthquakes) which could indicate the level of perceived risk." |
| 13 | 17-18 Please be more specific. Conclusions about what? Whether there is a risk of seismicity due to hydraulic fracturing?

We see your point. The text now reads "Using the UK as a case study, we examine the conclusions of expert-led public-facing reports on the risk (likelihood and impact) of seismicity induced by hydraulic fracturing for shale gas published between 2012 and 2018 and the terminology used in these reports" |
| 14 | 22 Add colon : after years

Colon added |
| 15 | 23-26 This is possibly the most important finding of your research, so please state it more explicitly, e.g., that what appeared to be polarization in their views was actually the result of different interpretations of the language used. "By examining the rationale provided for their answers we find that what appeared to be polarisation of views amongst experts was actually the result of different interpretations of the language used to describe seismicity." Responses are confounded by ambiguity of language around earthquake risk, magnitude, and scale.

Agree. Rephrased to: "By examining the rationale provided for their answers we find that an apparent polarisation of views amongst experts was actually the result of different interpretations of the language used to describe seismicity. Responses are confounded by ambiguity of language around earthquake risk, magnitude, and scale. We find that different terms are used in the survey responses to describe |

| | earthquakes, often in an attempt to express the risk (magnitude, shaking, potential for adverse impact) presented by the earthquake, but that these terms are poorly defined and ambiguous and do not translate into everyday language usage." |
|---|---|
| | **Highlights** |
| 16 | Green highlights indicate grammar/punctuation/spelling/redundancy issues. I am in the U.S.; it is possible that I have highlighted something that is standard for UK English, in which case, please ignore.

Thank you. All of these have been resolved. |
| 17 | Salmon highlights indicate a need for rewording or changes in punctuation due to ambiguity, confusing language, unnecessary language, or unnecessary parentheses or dashes.

Thank you. All of these have been resolved. We note the irony of using confusing language to present a paper about confusing language. |
| 18 | Yellow highlights indicate clarification needed.

Thank you. We provide clarification to all instances. |

---

## Author Response (AR3)

26th April 2021

Dear Editor,

I noted your comment that tables 5 and 6 cannot contain coloured cells due to HTML conversion of the paper. The coloured cells greatly help the reader to interpret the data and so we convert these to Figures and have updated/amended the text accordingly and provide them as figures in the associated ZIP file.

Thanks and best wishes,

Jen Roberts

Corresponding Author

**The place of useful learning**

The University of Strathclyde is a charitable body, registered in Scotland, number SC015263